# Immune phenotyping of diverse syngeneic murine brain tumors identifies immunologically distinct types

Jasneet Kaur Khalsa[1,2], Nina Cheng [1,2], Joshua Keegan[3], Ameen Chaudry[1], Joseph Driver[2], Wenya Linda Bi[2], James Lederer[3] & Khalid Shah [1,2,4✉]

Immunotherapy has emerged as a promising approach to treat cancer, however, its efficacy in highly malignant brain-tumors, glioblastomas (GBM), is limited. Here, we generate distinct imageable syngeneic mouse GBM-tumor models and utilize RNA-sequencing, CyTOF and correlative immunohistochemistry to assess immune-profiles in these models. We identify immunologically-inert and -active syngeneic-tumor types and show that inert tumors have an immune-suppressive phenotype with numerous exhausted CD8 T cells and resident macrophages; fewer eosinophils and SiglecF+ macrophages. To mimic the clinical-settings of first line of GBM-treatment, we show that tumor-resection invigorates an anti-tumor response via increasing T cells, activated microglia and SiglecF+ macrophages and decreasing resident macrophages. A comparative CyTOF analysis of resected-tumor samples from GBM-patients and mouse GBM-tumors show stark similarities in one of the mouse GBM-tumors tested. These findings guide informed choices for use of GBM models for immunotherapeutic interventions and offer a potential to facilitate immune-therapies in GBM patients.

[1] Center for Stem Cell Therapeutics and Imaging, Brigham and Women's Hospital, Harvard Medical School, Boston, MA 02115, USA. [2] Department of Neurosurgery, Brigham and Women's Hospital, Harvard Medical School, Boston, MA 02115, USA. [3] Department of Surgery, Brigham and Women's Hospital, Harvard Medical School, Boston, MA 02115, USA. [4] Harvard Stem Cell Institute, Harvard University, Cambridge, MA 02138, USA. ✉email: kshah@bwh.harvard.edu

Glioblastoma (GBM) is the most prevalent central nervous system (CNS) malignancy diagnosed in adults. Despite advances in surgical, chemoradio-therapeutic treatment regimens, the median overall survival for patients receiving the standard of care remains poor[1,2]. Immunotherapy has emerged as a promising approach for cancer[3] and achieved an unprecedented success in malignancies such as prostate cancer and melanoma[4,5]. However, its efficacy in GBM has been limited primarily by overall systemic immune suppression and the immune-suppressive tumor microenvironment[6,7]. A number of preclinical studies assessing the efficacy of different therapeutics for GBM have focused on patient derived tumor models with the caveat that these xenografts can only grow in immune-compromised mice limiting their use in gaining new insights into the mechanisms associated with brain tumor immunotherapy-based research. Preclinical syngeneic mouse tumor models play a critical role in testing and understanding the immune response of novel therapies prior to their clinical trial in patients. However, despite the wealth of available established and primary GBM lines to generate pathologically and genetically distinct xenograft models, the availability of syngeneic mouse tumor lines is limited and the models generated from these lines have not been well characterized.

Commonly used GBM syngeneic tumor models are p53 WT/ PTEN deficient CT2A[8] and K-ras mutant/p53 mutant GL261[9] lines. While both these tumor lines were generated by chemical induction with methylcholanthrene in C57BL/6 mice[10], there is a huge disparity in the number of studies performed on these two models. Most preclinical research for immunotherapy in GBM has been performed on GL261 which is highly immunogenic[11–13] resulting in outcomes that do not correlate well with the clinical findings[14]. Among other sporadically used syngeneic mouse tumor lines are spontaneous tumor models 005, which has an H-ras and AKT activation in a p53+/− setting[15], and Mut 3 and Mut 4 lines generated by inactivation of NF1 and p53 tumor suppressor genes with loss of PTEN[16].

To fully utilize the translational differences in various syngeneic mouse GBM models, it is crucial to characterize the tumor micro-environment. Based on TCGA (The Cancer Genome Atlas) data for gene expression profiling and mutational spectrum that revealed immune phenotypic differences in different GBM subtypes[17,18], we hypothesized that the tumor models generated from mouse tumor lines with distinct background would have differences in their immune profiles. This is further strengthened by previous findings showing that immunotherapy with autologous tumor lysates of GL261 and CT2A has a differential response with better efficacy in CT2A tumors[11].

Previously considered to be an immune-privileged site, it has been established that there is active surveillance by the immune cells in the brain[19]. Studies involving isolation of immune cells from the brain and flow cytometry-based evaluation of brain-infiltrating immune cells present unique challenges such as poor viability and high autofluorescence[20]. Recently, these challenges were overcome for brain tissue by utilizing better isolation protocols and flow cytometry to study up to 21 markers simultaneously. More recent advent of mass cytometry (CyTOF) has proven to be a powerful tool as analysis of up to 40 different markers can be performed from the same sample in a single experiment[21]. In addition, advances in analyzing and visualizing CyTOF data by unsupervised dimensionality reduction and subsequent clustering methods has remarkably enhanced our capabilities of meaningfully interpreting information of 40 or more different markers on millions of cells. CyTOF data, in combination with RNA sequencing (RNA seq) and correlative immunohistochemistry (IHC) provides a holistic overview of the leukocytic landscape of the currently available GBM tumor models and a resource to assist future immune-modulatory studies.

In this study, we seek to characterize the limited pool of diverse mouse GBM lines with pathological techniques, i.e., H&E staining, IHC, and RNA seq. Comprehensive immune profiling is done with CyTOF. Clinically, initial treatment for GBM patients is often resection, which as an intervention, is immunogenic resulting in an increase in tumor infiltrating immune cells (TIICs) in mouse GBM models[22,23]. In this study, we test the influence of resection on immune response, by performing immune profiling of tumors by CyTOF on pre- and post-resected tumors. Finally, we perform CyTOF on intraoperative tumor tissue obtained from GBM patients and compare it to our mouse GBM tumor models and report similarities between patient and mouse models with respect to immune phenotypes.

## Results

**Syngeneic mouse tumor models are genetically distinct and have different immunogenicity**. We sought to extensively characterize the differences between the commonly used syngeneic mouse GBM models by pathological characterization, RNA seq analysis, and immune phenotyping of tumors grown intracranially (Fig. 1a). Western blot analysis of lysates prepared from five lines: 005, CT2A, GL261, Mut3, and Mut4 confirmed that CT2A and GL261 have low levels of PTEN and high levels of p53 (Fig. 1b). On the contrary, 005 has high levels of PTEN and low p53 levels and p53 mutant Mut3 and Mut 4 both lacked PTEN[16] (Fig. 1b) with concomitantly high levels of p-AKT in Mut4 as compared to Mut3. To track tumor growth in vivo, we transduced syngeneic GBM lines to express dual imaging marker GFP and Firefly luciferase (Fluc); and created GBM-GFP-Fluc lines (Supplementary Fig. 1A, B). In vivo, four of the five lines formed tumors in immune competent C57BL/6 mice except for Mut4 (Supplementary Fig. 1C). Mice implanted with CT2A tumors took the shortest time to reach end-stage while tumor progression of Mut 3 was the slowest (Fig. 1c). H&E staining on brain sections from end-stage brains showed the presence of tumors (Fig. 1d) that was further confirmed by GFP expression in tumor cells in brain sections (Supplementary Fig. 1D). Four tumor models used showed varying degrees of angiogenesis as scored by CD31 staining (Fig. 1d). CD31 staining revealed that Mut3 tumors have significantly higher number of blood vessels as quantified by tube length (Supplementary Fig. 1E) correlating with its low-PTEN expression (Fig. 1b). PTEN levels are inversely correlated with angiogenesis levels[24]. Ki67 staining revealed presence of highly proliferating cells with high density of nuclear staining. Interestingly, GL261 and 005 tumor sections showed several non-proliferating cells in the tumor mass as compared to CT2A and Mut3 suggestive of infiltrating non-proliferating cells (Fig. 1d, Supplementary Fig. 1F).

Next, we performed RNA seq analysis to study the immune profile of different GBM tumor models. End-stage tumors were excised, and RNA was isolated from tumor-bearing or naïve brain tissue. Principal component analysis (PCA) of the data set revealed that PC1 and PC2 together comprise of more than 70% variance in the data (Supplementary Fig. 2A). There was more intergroup variability as PC1 could separate naïve brain from all the tumor samples whereas PC2 separated the Mut3 tumor suggesting that expression profile of Mut3 is dramatically different from the other three tumor samples (Supplementary Fig. 2A). Heatmap of clustered differentially expressed (DE) top 100 genes correlated with the PCA analysis. Clustering analysis of the heatmap confirmed that intergroup variability was higher than intragroup variability as different samples of the same group clustered together (Supplementary Fig. 2B), naïve brain sample being the most distinctive.

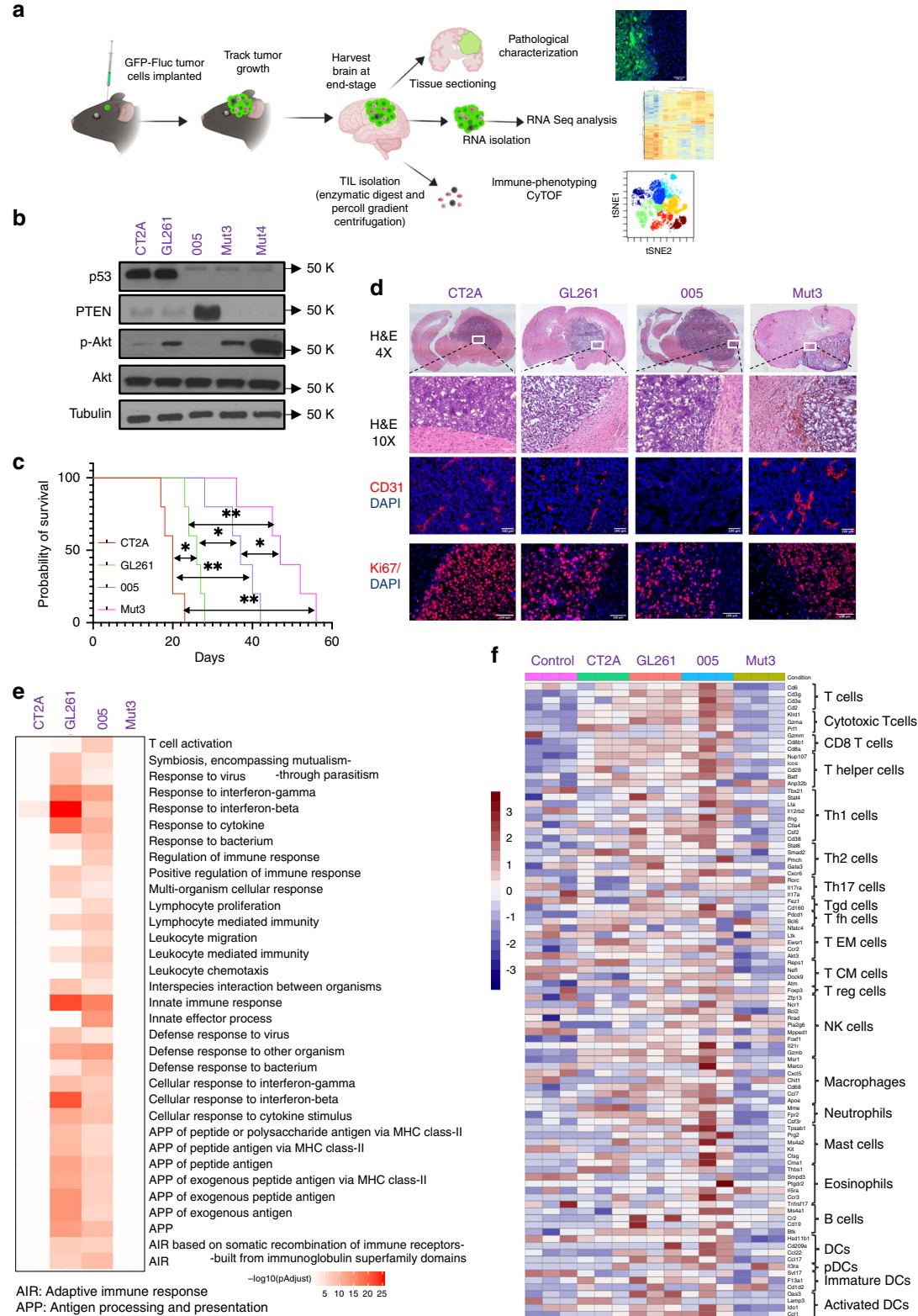

**Fig. 1 Characterization of syngeneic mouse brain tumor models. a** Schematic of the experimental plan. **b** Western blot analysis showing expression of p53, PTEN, pAKT, and AKT in the lysates prepared from mouse GBM syngeneic lines. α-Tubulin was used as loading control. Images are representative of five independent experiments. **c** Kaplan–Meier plots showing survival of various mouse tumor models ($n = 5$). Log-rank test was performed to compare each pair of survival curves. $*p \leq 0.05$; $**p \leq 0.005$. **d** H&E (magnification stitched 4× image and 10×), Ki67 and CD31 staining on the brain sections obtained from mice bearing tumors at end stage. Images are representative of three independent experiments (**e**) RNA sequencing analysis on the RNA isolated from end-stage tumors or control brains of C57BL/6 mice ($n = 3$ mice per group). Pathway enrichment analysis was performed, and the $p$-adjusted values were plotted for each tumor type for immune-related pathways that appeared in the top 25 enriched pathway list as shown in Supplementary Figs. 3 and 4. **f** Heatmap of differential expression of genes associated with immune cell types in different tumor types plotted as $z$-score of normalized gene expression for each gene. **a** was created by authors using biorender tools (biorender.com).

Pathway enrichment of DE genes, either upregulated or downregulated in each tumor compared to control naïve brain tissue revealed interesting findings relevant to immune-related pathway activation. Looking at the list of top 25 pathways most significantly differentially enriched in the DE genes from each tumor to naïve comparison, CT2A and Mut3 vs. naïve DE genes were enriched in pathways related to cell cycle processes and regulation that would be pertinent for a highly proliferating tumor (Supplementary Fig. 3). On the contrary, GL261 and 005 tumor to naïve comparison DE genes were enriched for genes from multiple pathways relevant to immune response suggesting that these tumors might have a higher frequency of immune cells in the tumor micro-environment and immune-activation pathways (Supplementary Fig. 4). Comparing the immune-related pathways significantly enriched in GL261 and 005 in all tumor types revealed that genes from immune response-related pathways were not significantly enriched in CT2A and Mut3 DE genes. In comparison, DE genes from GL261 and 005 tumors were highly enriched for genes related to immune pathways suggesting that GL261 and 005 are more immunologically active than CT2A and Mut3 tumors (Fig. 1e). Also, high immunogenicity correlates with low vascularity[25] as can be appreciated from low CD31 expression in 005 and GL261 tumors (Fig. 1d) and these two tumor types elicited more changes in immune response genes as seen from the pathway enrichment analysis (Fig. 1e). We further compared expression levels of immune-cell type-annotated genes as previously described[26]. Mut3 tumors showed lowest expression of immune-cell related genes while 005 and GL261 tumor types were enriched in immune cells specifically T cells, macrophages and eosinophil related genes (Fig. 1f). These findings reveal that syngeneic GBM mouse tumors have variability in their immune profiles independent of whether tumor lines were generated by chemical mutagenesis or derived from spontaneous tumors.

**Tumor-bearing brains have a substantially different immune cell profile than naïve brain.** To extensively characterize the differences in immune profiles, we utilized CyTOF that allows simultaneous staining of cells isolated from tumor tissue/naïve brain with 31 metal-conjugated antibodies[21]. We confirmed CyTOF staining by flow cytometry[27] and found no difference in proportions of CD4/CD8 and Tim3+ CD4/CD8 T cells in mouse tumor samples (Supplementary Fig. 5). The antibody panel for analyzing mouse tumors by CyTOF includes markers to identify both adaptive and innate immune cells along with activation/inhibition markers (Supplementary Fig. 6A). We first sought to establish differences in immune cell population in the most commonly used GBM mouse model, GL261, tumor-bearing brain in comparison with naïve brain tissue. CyTOF data files were analyzed for FlowSOM clustering as outlined in Supplementary Fig. 6B–D and described in the "Methods" section. FlowSOM run with hierarchical consensus clustering produced optimal immune subset islands when set at seven metaclusters, revealing distinct spatially separated populations of immune cells in naïve brain and tumor-bearing brain (Fig. 2a). As shown previously[19], resident immune cells in the naïve brain sample had low expression of CD45, low-CD11b expression and high expression of CX3CR1 (Fig. 2b). On the other hand, the heatmap shows that tumor-bearing samples showed higher CD45 expression in all of its populations (Fig. 2c). Interestingly, some populations of infiltrating immune cells in the tumor also showed high levels of CX3CR1 expression (Fig. 2b, c). To define metaclusters, we visualized the individual marker expression levels of the previously defined seven metaclusters in both naïve and tumor-bearing samples (Fig. 2c). Three metaclusters that have low-CD45

expression (metacluster 4–6) that could be microglial subsets were identified. All of these have high CX3CR1 expression[28]: metacluster 4 has high CD11c and CX3CR1, defined as CD11c+ microglia; metacluster 5 has CD115 expression as well, defined as CD115+ microglia. Ly6C expression was seen in metacluster 6, however with low CD45 expression and therefore defined as Ly6C+ microglia/monocytes.

Metacluster 1 (CD4 cells), 2 (macrophages) and 3 (CD8/B cells) showed significantly higher frequency in tumor-bearing sample while metacluster 4 (CD11c+ microglia) and 5 (CD115+ microglia) showed significantly higher frequency in naïve brain samples (Fig. 2d). Macrophage/monocytes (metacluster 2) constitute the highest frequency of immune cells in the tumor-bearing tissue in accordance with patient data[29,30]. On the other hand, bulk of immune cells in naïve brain belong to cluster 4 and 6 representing different microglial subpopulations. A few microglial cells present in the tumor-bearing sample showed higher expression level of MHCII and CD45 levels as compared to microglial cells in the naïve brain (Fig. 2c). We used biaxial gating to define microglial cells as CD11b mid/CD45 low followed by gating on MHCII and CX3CR1 to identify resting and activated microglia (Fig. 2e)[31,32]. To check whether other tumor types also have the same trend for activated and resting microglial cells, we performed similar analysis on all the 4 different tumor types (Supplementary Fig. 7). Each tumor type had <10% of resting microglia in comparison to >50% in naïve brain. Conversely, naïve brain has <10% of activated microglia and all tumors had between 35 and 55% activated microglial cells. Tumor-bearing brain has a majority of infiltrating immune cells and few microglial cells. These few microglial cells also have an activated phenotype (Supplementary Fig. 7). These findings reveal that the immune-phenotype of a tumor-bearing brain is in stark contrast with the naïve brain[19] thus allowing us to assess differences in immune phenotypes in syngeneic mouse models.

**Baseline immune profiling of different syngeneic mouse GBM models.** Next, to compare the immune profiles of 4 different syngeneic mouse models, we applied a similar analysis workflow as described above and ran a flowSOM analysis with 25 clusters. As the number of TIICs isolated from various tumor models was quite different with variability in the CD45+ population, we therefore focused our study on the composition of the TIICs rather than absolute cell numbers. All samples of the same tumor type were concatenated after FlowSOM analysis and individual marker expression was used to define 21 distinct populations(Fig. 3a). A small population of cells was left undefined (called unknown) due to lack of population defining marker expression. CT2A had fewer activated (CD45 low CD11b low CX3CR1+ MHCII+) and resting microglia (CD45 low CD11b low CX3CR1+ MHCII−) than 005 tumors (Fig. 3b). GL261 tumors had higher frequency of activated microglia than CT2A tumors (Fig. 3b). For the innate immune cell populations, Mut 3 tumors had less than half the frequency of resident macrophages (CD11b+ F4/80+ CD64+ Ly6C−) than CT2A tumors (Fig. 3c). Furthermore, Mut 3 had significantly fewer SiglecF+ macrophages (SiglecF+ CD11b+ MHCII+ F4/80+) and type A DCs (CD11c++ CD11b+ MHCII−) than 005 tumors (Fig. 3c).

T cells make an important component of the antitumor immune cell brigade. We identified three subsets of CD4 T cells: classical CD4 (CD3+ CD4+ CD44+ MHCII+ Tim3− Lag3− CD25−), exhausted CD4 (CD3+ CD4+ Tim3+ Lag3+) and CD4 T regs (CD4+ CD25+ KLRG1+ CD103+). Interestingly, there was similar frequency of exhausted CD4 T cells in all tumor types (Fig. 3d). Mut3 tumors had more classical CD4 T cells than 005 tumors. Mut3 also had significantly more regulatory T cells

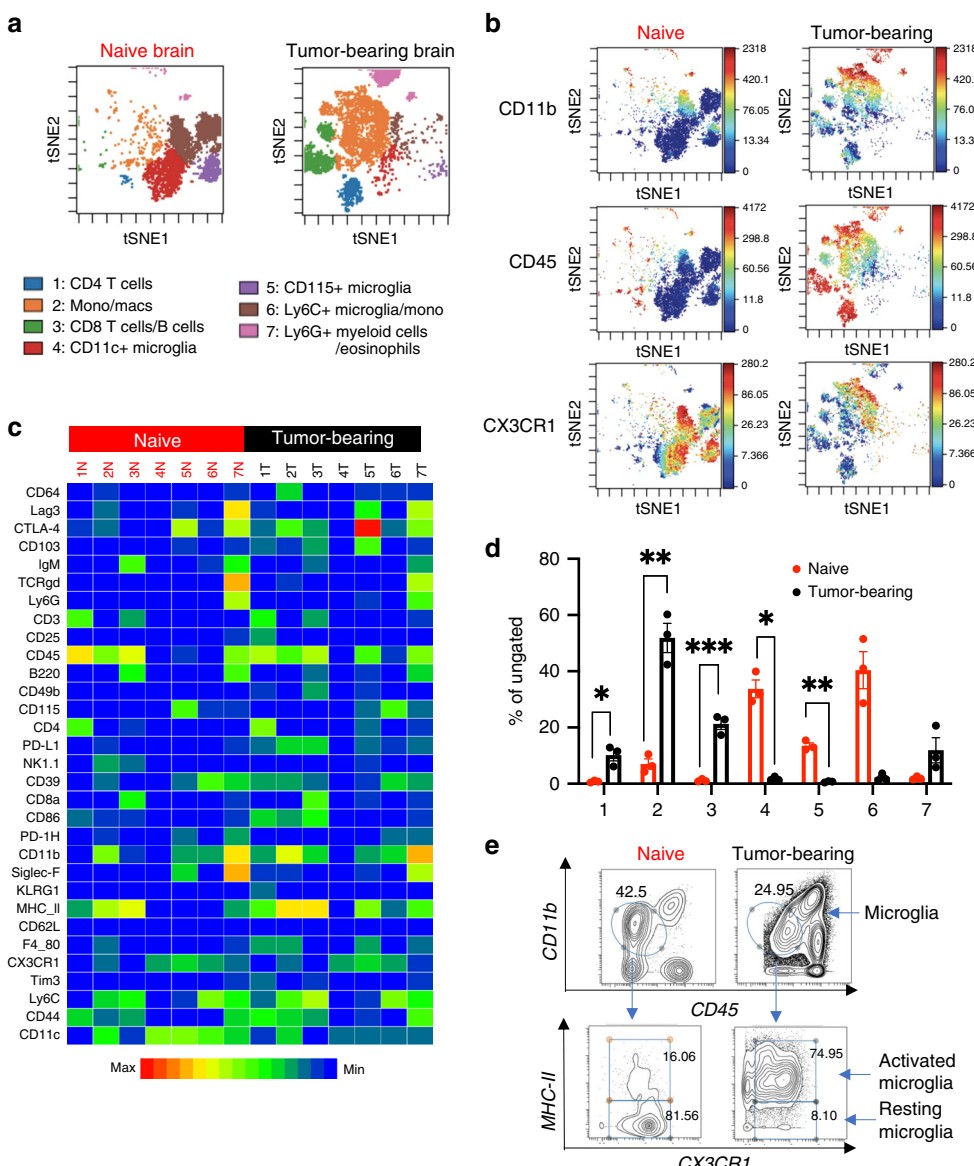

**Fig. 2 Tumor-bearing brain shows distinct population of infiltrating immune cells. a** Concatenated FlowSOM analysis of naïve brain vs. tumor-bearing brain showing seven metaclusters (representative plot from two independent experiments with n = 3 mice/group). **b** Expression of CD11b, CD45 and CX3CR1 in viSNE clusters of naïve vs. tumor-bearing brain. **c** Heatmap showing expression profile of metaclusters in concatenated samples. **d** Percent population in each metacluster of naïve vs. tumor-bearing brain. Data represented as average ± SE for n = 3 mice/group. Data representative of two independent experiments. Two-sided Student's t test with Holm−Sidak corrections for multiple comparisons was applied. *p ≤ 0.05; **p ≤ 0.005; ***p ≤ 0.0007. **e** Representative flow plot of resting and activated microglia in naïve vs. tumor-bearing brain. Microglia was gated as CD11b low/CD45 low which was further gated for MHC-II high/CX3CR1 high (activated microglia) and MHC-II low/CX3CR1 high (resting microglia).

than other tumors tested (Fig. 3d). We also identified three CD8 T cell subsets: exhausted CD8 (CD39+ Tim3+ Lag3+ CD8+), classical CD8 (CD3+ CD8+ CD44+ MHCII+ Tim3− Lag3− CD25−) and tumor-reactive CD8 (CD39+ CD103+ CD8+). Mut3 tumors also had a higher frequency of classical CD8 T cells than GL261 and 005 tumors, while CT2A had the highest exhausted CD8 T cell frequencies (Fig. 3d).

We also confirmed population clustering by utilizing the more traditional and complementary data processing approaches on GL261 tumors: SPADE and viSNE. SPADE clusters phenotypically similar cells in a hierarchy[33]. We first assessed the expression of markers that are commonly used to define various myeloid and lymphoid immune-cell populations as two-dimensional dot plots and their expression on the SPADE plot (Supplementary Fig. 8). The two subsets of T cells: CD4 (CD45+

CD4+) and CD8 cells (CD8+ CD45+) diverge from a common node and form distinct clusters. Similarly, IgM+ B cells (B220+ IgM+), NK cells (NK1.1+ CD49b+), monocytes/macrophages (CD11b+ CD64+), SiglecF+ myeloid cells (SiglecF+ CD45+), Ly6G+ myeloid cells (Ly6G+ CD11b+) and dendritic cells (MHCII+ CD11c+) form distinct clusters on the SPADE plot. Microglia cells were gated as CX3CR1+ CD45 low in the biaxial dot plot and these formed a cluster that had a low-CD11b expression as can be seen on the SPADE plot.

Expression of various markers in GL261 tumors (Supplementary Fig. 5A) were mapped on viSNE plot (Supplementary Fig. 9). Most markers form distinct clusters based on their expression level. When CD39 expression was compared on viSNE plots of concatenated files for each tumor type (Supplementary Fig. 10A), all tumor samples showed high CD39 expression that mapped to

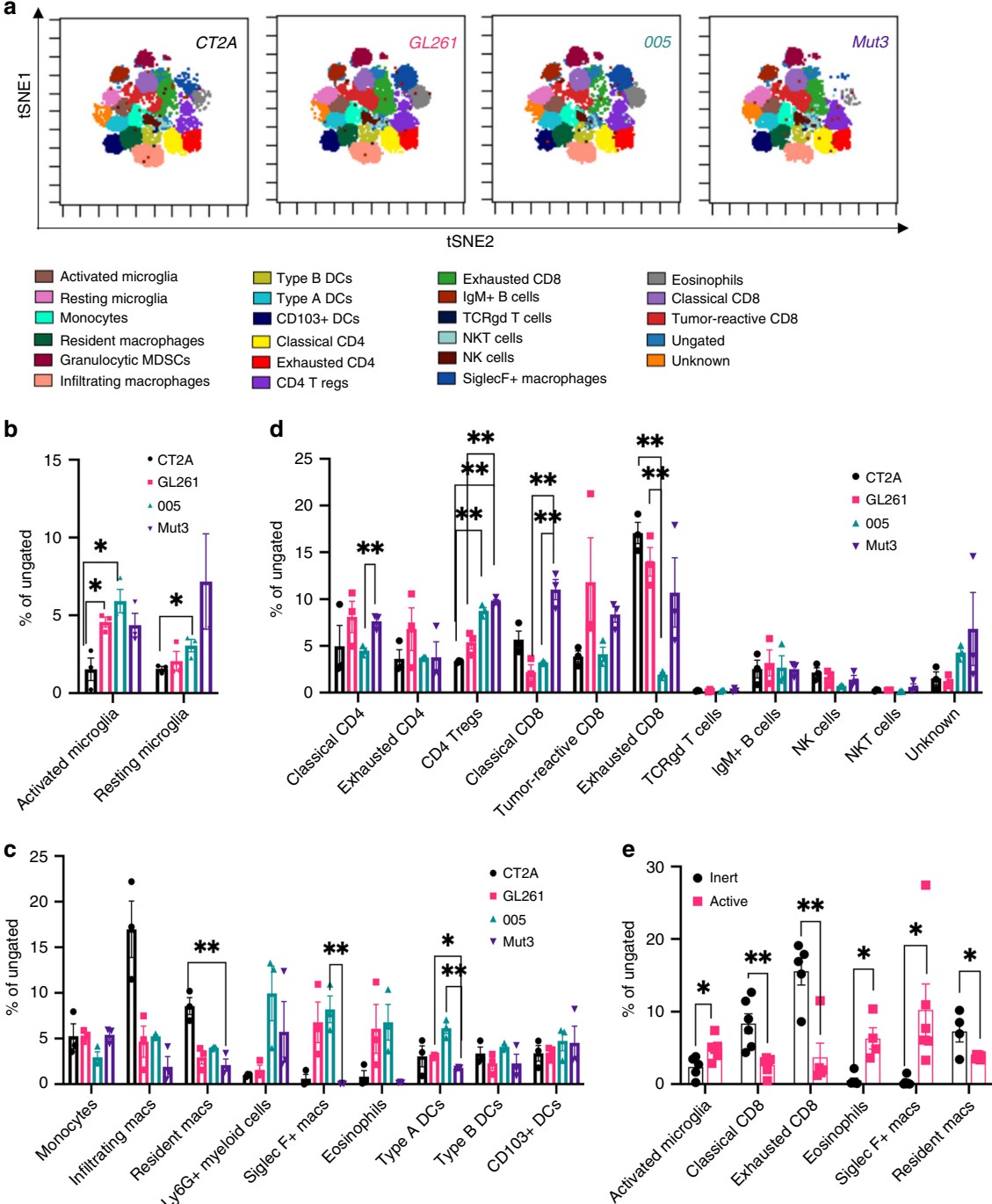

**Fig. 3 FlowSOM analysis on immune cells in syngeneic mouse tumors. a** Concatenated FlowSOM plot of different syngeneic tumor mouse models showing various populations defined based on expression analysis. Major markers used for defining clusters are activated microglia (CD45 low CD11b low CX3CR1+ MHCII+), resting microglia (CD45 low CD11b low CX3CR1+ MHCII−), monocytes (CD11b+ F4/80− CD64+ CX3CR1+ CD11c−), resident macrophages (CD11b+ F4/80+ CD64+ Ly6C−), Ly6G+ myeloid cells (Ly6G+ Ly6C+ CD11b+), infiltrating macrophages (CD11b+ F4/80+ CD64+ Ly6C+), Type B DCs (CD11c++ CD11b+ MHCII+), Type A DCs (CD11c++ CD11b+ MHCII−), CD103+ DCs (CD103+ CD11c+), classical CD4 (CD3+ CD4+ CD44+ MHCII+ Tim3− Lag3− CD25−), exhausted CD4 (CD3+ CD4+ Tim3+ Lag3+), CD4 T regs (CD4+ CD25+ KLRG1+ CD103+), exhausted CD8 (CD39+ Tim3+ Lag3+ CD8+), IgM+ B cells (B220+ IgM+), TCRgd T cells (TCRgd+ CD3+), NKT cells, (NK1.1+ CD49b+ CD3+ CD8+), NK cells (NK1.1+ CD49b+ CD3−), SiglecF+ Macrophages (SiglecF+ CD11b+ MHCII+ F4/80+), eosinophils (SiglecF+ CD11b+ MHCII−), classical CD8 (CD3+ CD8+ CD44+ MHCII+ Tim3− Lag3− CD25−), tumor-reactive CD8 (CD39+ CD103+ CD8+). **b–d** Abundance analysis on microglial clusters, innate immune cells, and adaptive immune cells in four tumor types as defined in **a**. Data represented as average ± SE for n = 3 mice/group; Representative data from two independent experiments. **e** Tumors were classified into immunologically active (GL261 and 005) and immunologically silent (CT2A and Mut3) based on RNAseq analysis. FlowSOM analysis was performed followed by abundance analysis. Populations that were significantly different were plotted. Data is represented as average ± SE for each cluster characterized. Two-sided Student's t test with Holm–Sidak corrections for multiple comparisons was applied. *p ≤ 0.05; **p ≤ 0.005.

CD4 and CD8 T cells (Supplementary Fig. 9). CD39 is an ecto-ATPase that inhibits immune responses and CT2A tumors have the overall highest mean CD39 expression level, specifically in CD8 T cells (Supplementary Fig. 10B, C).

Next, we evaluated CD4 and CD8 T cells for expression of various activation and inhibition markers (Supplementary Fig. 11A, B). GL261 tumors showed the lowest frequency of KLRG1+ cells and CD25+ cells of the CD4 subset (Supplementary Fig. 11A), whereas 70–80% of CD8 T cells in CT2A were positive for Lag3 and Tim3, markers of dysfunctional T cells in comparison to the other tumor types analyzed (Supplementary Fig. 11A). Also, the percentage of PD-1H+CD4+ was higher than PD-1H+ CD8+ T cells in all tumor types as previously reported[34].

To quantify various immune-cell populations, we performed correlative IHC (Supplementary Fig. 12). CT2A tumors showed fewer total CD3+ T cells as compared to GL261 and 005 tumors. Mut3 tumors had fewer CD4 and CD8 T cells than GL261 tumors. The number of Foxp3+ cells was more than CD4 T cells suggesting that Foxp3 is expressed in other cell populations, possibly CD8 T cells, as well. These findings correlate with lowest percentage of KLRG1+ CD4+ T cells in GL261 tumors (Supplementary Fig. 11A). CD68+ macrophages showed similar frequencies in all the four tumor types. 005 tumors had the highest number of Iba-1+ cells that scores for microglia. Overall, CT2A and Mut3 tumor types have higher frequencies of inhibitory and exhausted immune cells in the tumor microenvironment with CT2A showing fewer total T cells, high levels of CD39, fewer activated microglia and exhausted CD8 T cells, while Mut3 had fewer CD4/CD8 T cells characterized by higher frequency of regulatory CD4 T cells and fewer type A DCs.

**Immunologically inert tumors have an immune-suppressive phenotype**. Based on RNA sequencing data (Fig. 1d), we classified CT2A and Mut3 tumor samples as immunologically inert and GL261 and 005 tumors as immune-active tumors. Abundance analysis was performed on combined samples for immunologically inert and active tumors (Fig. 3e, Supplementary Fig. 13A). Immunologically active tumors showed lower proportions of exhausted CD8 T cells, classical CD8 T Cells and resident macrophages and higher proportions of eosinophils, SiglecF+ macrophages and activated microglia (Fig. 3e). When proportions of activated and suppressive markers in CD4 and CD8+ T cell population were compared, immunologically active tumors had fewer Tim3+ and CD39+ CD8 T cells (Supplementary Fig. 13B, C). In CD4+ T cell subset, PD-1H+ T cells (exhausted T cells) were fewer in immunologically active tumors, while CD103+ CD4+ T cells (resident memory T cells) were higher in immunologically active tumors. These data, together with RNA seq data, indicate that immunologically inert tumors have fewer infiltrating immune cells and the phenotype of these cells is suppressive.

**Resection invigorates an anti-tumor immune response**. As tumor debulking is the first line of treatment for GBM patients, we explored whether surgical intervention in mouse GBM tumors would result in a differential immune response. We assessed the immune phenotype of an immunologically inert CT2A tumor 4 days post tumor-resection in comparison to an unresected tumor. CT2A tumors were superficially implanted in cranial window of C57BL/6 mice (Fig. 4a). Tumor growth was tracked by bioluminescent imaging (Fig. 4b) and 14 days post-tumor implantation, fluorescent-guided tumor resection was performed and tumors were harvested 4 days post-resection. FlowSOM cluster analysis on concatenated resected samples showed

an increase in T cells and activated microglia post-tumor resection (Fig. 4c). Comparative abundance analysis of the frequencies of broad immune populations showed an increase in activated microglia, CD4+ and CD8+ T cells, and SiglecF+ macrophages with a concomitant decrease in frequencies of resting microglia and resident macrophages (Fig. 4d, Supplementary Fig. 14).

Mean expression levels of CD39, an inhibitory marker, revealed a reduction in both CD4 and CD8 T cells (Fig. 4e). Similarly, there was a reduction in PD-1H expression in CD8 T cells and an increase in CD103 and Tim3 post-resection. Previous studies have shown that Tim3 is upregulated on short-lived effector T cells[35] and CD103 is also a marker for tumor-reactive CD8 T cells. We observed lower PD-L1 and higher CD25 expression levels in CD4 T cells post-tumor resection (Fig. 4e). These data suggest that tumor resection removes GBM tumor core that has suppressive immune cells while concomitantly promoting infiltration of new immune cells that have an activated phenotype.

**Tumor infiltrating immune cell profiling of GBM patients resembles 005 tumors**. In order to recapitulate clinical findings for immune-modulatory therapies, the baseline immune profile of syngeneic mouse models should be similar to that of untreated patients. We compared the immune profile of our syngeneic mouse models with patient tumor immune profile. Clinical pathology report on tissue harvested from WHO grade-IV GBM patients indicated that all the tumors had an IDH-WT genotype with equal representation of MGMT methylated and non-methylated. Patient-derived GBM samples showed a high proportion of antigen presenting cells amounting to a total of 60% that include resting and activated microglia, infiltrating, and resident monocytes/macrophages (Fig. 5a, b; Supplementary Fig. 15). Very few B cells and NK cells and a total of 20% T cells were present in GBM patient samples. Statistical analysis performed to compare each of the mouse tumor model to patient immune-phenotyping data revealed the presence of significantly fewer dendritic cells in patient samples as compared to any of the mouse tumor models (Fig. 5b). However, regulatory CD4 T cells frequencies were quite similar in mouse models and GBM patient samples. We identified CD103+ CD8 T cells, defined as tissue-resident memory T cells[36,37] in patient GBM samples (Fig. 5b) and tested the composition of matched patient blood for T-cell subsets. Performing FlowSOM analysis and comparing GBM tumor tissue, matched blood from patients and healthy donors showed distinct clusters present in the tumor tissue as compared to blood samples that corresponded to microglia, resident macrophages and eosinophils (Supplementary Fig. 16A). Abundance analysis of CD3+ T cells revealed that tumors had half the frequency of CD4 T cells than patient blood and CD4 T cells constituted almost 75% of the T cells in the blood of healthy donors as previously reported[38]. Furthermore, tumor infiltrating T cells had higher CD103+ CD8 T cells than found in blood of GBM patients (Supplementary Fig. 16B). These data indicate that the TIIC phenotype of GBM patients substantially varies from blood lymphocytes of these patients. To test which mouse tumor model resembles patient data most closely, we calculated the Spearman's rank correlation coefficient between patient data and each of the mouse tumor models that showed highest correlation was found in the 005 mouse tumor model and the lowest in the and CT2A tumor model (Fig. 5c). 005 tumor model might serve as a good preclinical model as it most closely resembles the immune-phenotypic signature of GBM patients.

## Discussion

In this study, we identified various specific immune cell populations in the highly malignant brain tumor microenvironment by

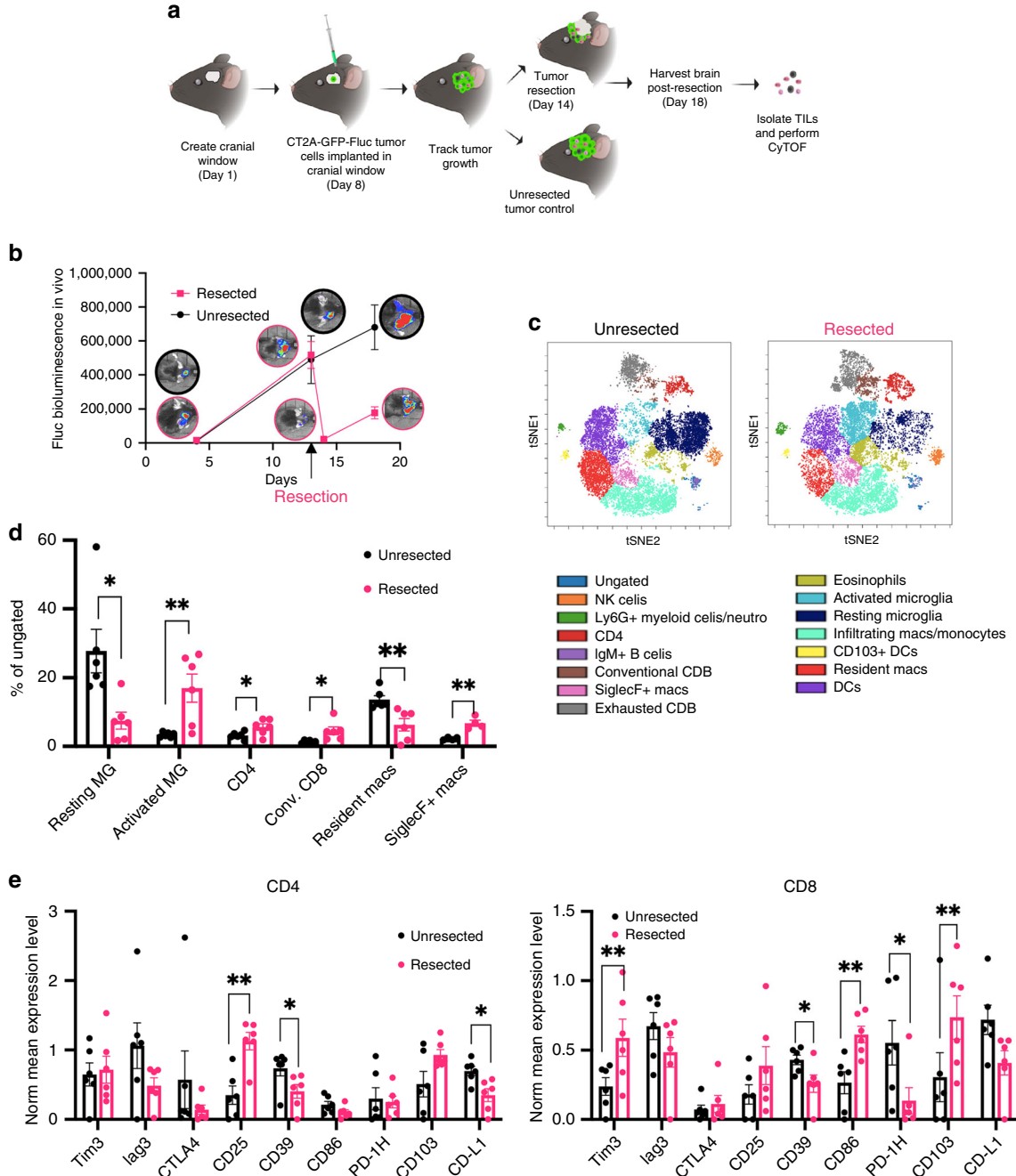

**Fig. 4 Tumor resection invigorates an anti-tumor immune response. a** Schematic of the experimental plan. **b** C57Bl/6 mice were implanted with $2 \times 10^5$ CT2A tumor cells in the cranial window. On day 14, tumors were resected using fluorescent guided microscopy. Tumor volume was monitored using bioluminescent imaging and plotted against time. Data represents average ± SE. $n = 6$ mice/group for two independent experiments. **c** FlowSOM analysis of tumor tissue collected 4 days post tumor-resection or unresected CT2A showing various populations defined based on expression analysis on various clusters. Populations were defined as following: NK cells (NK1.1+, CD49b+), Ly6G+ myeloid cells/neutrophils (Ly6C+/Ly6C+/CD11b+/CD45+), CD4+ (CD3+/CD4+/CD45+), IgM+ B cells (B220+ IgM+), Conventional CD8+ (CD3+/CD8+/CD45+), SiglecF+ macs (SiglecF+ CD45+ MHC-II+) exhausted CD8 (CD39+ Tim3+ Lag3+ CD8+), eosinophils (SiglecF+ CD45+ MHCII-ve), resting microglia (CD11b low/CD45 low/MHCII low), activated microglia (CD11b low/CD45 low/MHCII high), infiltrating macrophages/monocytes (CD11b+ F4/80+ CD64+ Ly6C+), CD103+ DCs (CD103+ CD11c+), DCs (CD11c+ MHCII+). **d** % of cells for populations showing significant differences were plotted as average ± SE ($n = 6$ mice/group, representative plot from two independent experiments). **e** Normalized mean expression values for various activation/inhibition markers in CD4 and CD8 T-cell populations in resected and unresected tumor samples were plotted as average ± SE ($n = 6$ mice/group, representative plot from two independent experiments). Two-sided Student's $t$ test with Holm–Sidak corrections for multiple comparisons was applied. *$p \leq 0.05$; **$p \leq 0.005$. **a** was created by authors using biorender tools (biorender.com).

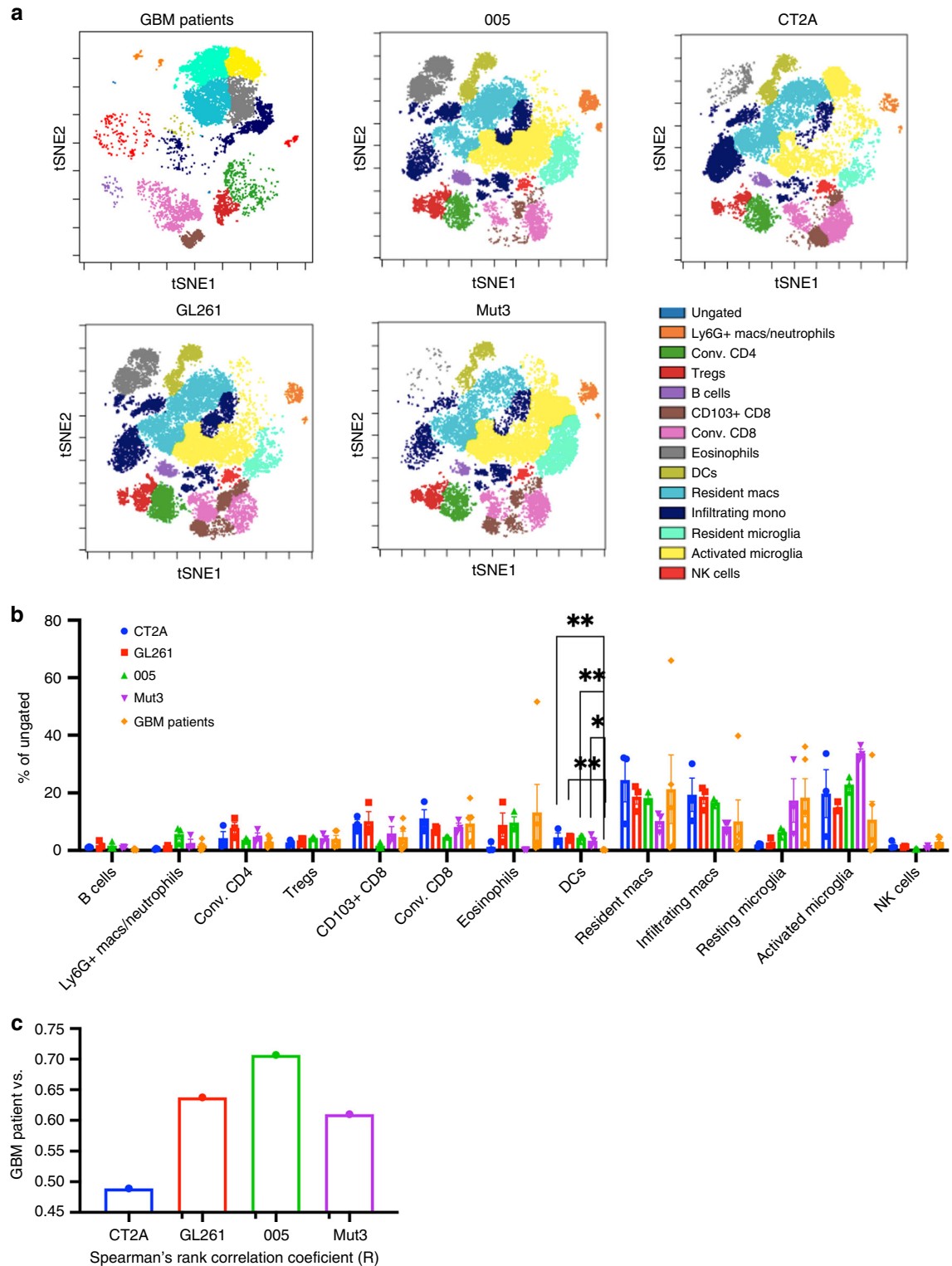

combining the power of RNA sequencing and CyTOF. Our findings show differences in immune-cell subsets in four syngeneic mouse tumor models allowing us to identify immunologically inert and active tumor types. CyTOF analysis revealed that the identity of immune cells and their activation status was different in the immunologically active vs. inert tumors. Finally, we show that immunologically inert GBM tumors in mice can be activated by surgical intervention, a standard of care procedure for GBM patients.

GBM xenograft models play a crucial role in evaluating efficacy and toxicity of new therapeutic drugs that directly inhibit tumor cell proliferation and survival. However, patient derived cells do not develop into tumors in immunocompetent mice and therefore are not suitable for studying the efficacy of any treatment modality that involves participation of immune cells. Immune-based therapies are showing promising results for many tumor types while assessing their efficacy for GBM is still in its infancy[2,3]. To further our understanding of the immune response

**Fig. 5 Patient tumor profile closely matches 005 mouse tumor. a** Population analysis of cells isolated from tumor tissue excised during surgery from GBM patients and mouse tumor samples as previously described. Murine populations were defined as following: B cells (B220+ CD45+), Ly6G+ Macs/ Neutrophil (Ly6C+ Ly6G+ CD11b+ CD45+), conventional CD4 (CD3+ CD4+ Tim3− Lag3− CD25− CD45+), regulatory T cells (CD4+ CD25+ KLRG1+ CD103+), CD103+ CD8 (CD3+ CD45+ CD8+ CD103+), conventional CD8+ (CD3+ CD8+ CD45+), eosinophils (SiglecF+ CD11b+ MHCII−), DCs (CD11c+ CD64+ CD86+ MHCII+ CD45+), resident macrophages (CD45+ CD11b+ MHCII+ Ly6C−), infiltrating macrophages (CD45+ CD11b+ MHCII+ Ly6C+), resting microglia (CD11b low CD45 low MHC-II low), activated microglia (CD11b low CD45 low MHC-II high), NK cells (NK1.1+ CD49b+). GBM patient populations were defined as B cells (CD19+ CD20+), Ly6G+ Macs/Neutrophil (CD15, CD66b), conventional CD4 (CD3+ CD4+ CD25− CTLA4−), regulatory T cells (CD4+ CD25+ CTLA4+) CD103+ CD8 (CD3+ CD45+ CD8+ CD103+), conventional CD8+ (CD3+ CD8+ CD45+), eosinophils (Siglec8+, CD68+), DCs (CD11c++ MHCII++ CD45+), resident macrophages (CD45+ CD11b+ CD14+ CD68+ CD172ab+ CCR2−), infiltrating macrophages (CD45+ CD11b+ CD14+ CD68+ CD172ab+ CCR2+), resting microglia (CD11b low CD45 low MHC-II low), activated microglia (CD11b low CD45 low MHC-II high), NK cells (NKp30+, CD16+). **b** Abundance analysis plotted as % of cells for populations defined in **a**. Data represented as average ± SE ($n = 3$ mice/group and five patient samples/group; two independent experiments). Two-sided Student's $t$ test with Holm–Sidak corrections for multiple comparisons was applied. *$p \leq 0.05$, **$p \leq 0.005$. **c** Spearman's rank correlation coefficient (R) was calculated for GBM patients with each of the mouse tumor models for $n = 3$ mice/group and five patient samples. Data representative of two independent experiments.

to the tumor and test different combinations of treatment in vivo, currently available syngeneic mouse GBM models provide a critical preclinical resource. Orthotopic syngeneic mouse GBM can be transplanted in immune-competent mice to generate tumors[39]. In this study, we used syngeneic imageable murine tumors which can be readily established and are highly reproducible. We utilized tumor cells from two chemically induced tumor models, CT2A and GL261, and three GEMM-derived cell lines, 005, Mut3, and Mut4 that are genetically distinct. These models, with the exception of Mut4, could form tumors when implanted intracranially in mice. Mice reached end-stage and showed neurological symptoms at varying time points post-implantation of different GBM lines.

With the advent of immunotherapy, it is essential to understand the various components of the tumor micro-environment of which both adaptive and innate immune cells are important players. Current approaches for evaluation of brain tumors is IHC where only 2–3 markers can be stained at the same time. RNA sequencing analysis of the bulk tumor tissue is common with the caveat that cellular identities cannot be defined[40]. Single-cell RNA sequencing is an alternative; however, cost and time are limiting factors. Flow cytometry with isolated TIICs is the most reliable method for immune-phenotyping and has been used frequently for identifying immune cell populations and their activation status. One of the major limitations with studying immune-cell subsets in brain tumors by regular flow cytometry is the limited number of fluorescent channels that can be analyzed at a time that gets further complicated by compensation issues resulting in multiple separate antibody cocktails[41]. As the number of TIICs isolated from the brain is low which limits the ability of flow cytometry to phenotype, CyTOF provides an excellent alternative as a large number of immune-cell population defining markers can be analyzed simultaneous, without any drop off in signal sensitivity[21]. We for the first time combined CyTOF and the more classical approaches of RNA sequencing and IHC to compare syngeneic mouse GBM tumors and provided a comprehensive overview of differences in TIIC phenotype and abundance.

Presence of immune cells in the naïve brain and their phenotype has been recently identified[42], however a comparison of immune cells present in the naïve brain versus tumor-bearing brain has not been explored. Our FlowSOM metacluster analysis showed that not only the infiltrating lymphocytes in the tumor-bearing mice cluster separately, but the phenotype of the existing immune cells in the brain, i.e., the microglia change significantly. Naïve brain has resting microglia while microglia in tumor-bearing brain develop an activated phenotype with an upregulation in their MHC-II expression, which is a marker of activation under inflammatory conditions[31].

Similar to GBM patients, our mouse models also have more than half of their TIIC constituted of TAMs while T cells are the major lymphoid cell population[43,44]. Tumor reactive CD39+ CD103+ CD8+ T cells were previously identified in ovarian, rectal, and melanoma patients, and correlated with increased overall survival[45]. We found this population in our syngeneic mouse models with the highest frequency in the GL261 mouse model. Their implication and function in mouse models and patients of GBM needs to be further analyzed.

One key difference that we found between the immune active and silent tumors is eosinophils. Tumor-associated eosinophilia have been associated with many human cancers, including GBM where eosinophilia is correlated with necrosis[46]. Eosinophils play an important role in various phases of GBM tumor growth: inhibiting initiation of GBM, triggering antitumor response during tumor promotion and slowing down progression[47]. Pretreatment eosinophil numbers act as a prognostic biomarker for survival[48] and a recent study in GBM patients showed that temozolomide treatment post-surgery induced eosinophilia and it was associated with improved survival[49]. Eosinophils also enhance T-cell homing and their role in macrophage polarization has been identified[50]. Our data indicate that mouse tumors with a lower frequency of eosinophils have an overall immune-suppressive microenvironment[37,51,52]. As we begin to understand the immune-regulatory role of eosinophils in other diseased conditions, their immune-modulatory functions in the context of GBMs need to be defined.

Tumor resection is the primary intervention for GBM patients before any other therapy is administered. Thus, it is imperative to understand the consequences of resection on immune-phenotype of the tumor to optimally administer different modalities of immune-therapy that would further modulate the immune response. Our data indicate that tumor resection in an otherwise immunologically silent tumor model, CT2A, results in an increase in T cells and SiglecF+ macrophage infiltration, while resident macrophages decline. Our data also indicate that recruited T cells post-tumor resection also have a more activated phenotype as has been previously reported in other tumor models[53,54]. Interestingly, immunologically active tumor types are comprised of more activated microglia and SiglecF+ macrophages along with fewer resident macrophages (Fig. 3e); suggesting that immune profile of an immunologically inert CT2A tumor post-resection correlates with that of an immunologically active tumor type. The focus of immune therapy approaches would need a paradigm shift to accommodate the modulated immune response which is more activated post-surgical intervention.

Finally, we compared GBM patient data to identify which of the GBM mouse models utilized in this study would reflect patient data most closely. Microglia frequency in GBM patients was between 30 and 50%, which corroborates with published literature[55]. Relative frequencies of T cells and macrophages infiltrating the tumor dictate the efficacy of various immune-modulatory

therapies[56] and a close resemblance in preclinical mouse models could better predict response to immunotherapy in patients. Although GL261 is widely used in preclinical studies, these tumor-bearing mice have significantly fewer APCs and more T cells than GBM patients that are usually suppressive in nature. This immune-phenotype could be a factor contributing to better efficacy of treatment modalities that target T-cell populations in GL261 in comparison to other mouse models[12]. In this study, we report a plethora of immune-phenotypic data that can be utilized to make better choices of the mouse model to be used for testing various antitumor therapies. We report higher frequency of Ly6C+ macrophages in CT2A that mediate efferocytosis and cross presentation of antigens that can explain why CT2A responds better to vaccination with autologous lysate, as compared to GL261 tumors[57].

Interestingly, tumor growth rate does not correlate with immune-suppressive or immune-inertness of the syngeneic mouse models tested. Mut3 is the slowest growing tumor model and takes the longest to reach end-stage, however based on Spearman's correlation co-efficient, it does not match well with the patient sample. Thus, 005 might serve as a better mouse model when immune modulatory therapies are to be tested. As all of our GBM patient samples are wildtype WHO grade-IV glioma, comparison of immune phenotypes of mouse models with patient samples from other GBM subtypes might yield different results.

We also phenotyped GBM tumor tissue with matched patient blood and included healthy donor blood for comparison. We observed that tumors have drastically different immune cell profile as compared to blood from patients or healthy donors and our findings are in concordance with the previous studies[38]. Furthermore, T cell subset analysis shows higher levels of CD8 T cells are more in the tumor, with respect to matched blood. Also, and CD103+ CD8 T cells are exclusively present in the tumor. CD103+ T cells define resident memory T cells that directly control tumor growth and presence of CD103+ CD8 T cells has been shown to correlate with better clinical outcome in lung, ovarian, cervical and breast cancer[58–60]. A deeper understanding of the role of CD103+ CD8 T cells and its effect on clinical outcome in GBM patients can significantly benefit patient survival.

We conclude that the mouse tumor model used in preclinical studies to test different immune based therapies should be carefully considered as their baseline immune state is variable. Further, for solid tumors like GBM where resection is the first line of treatment, testing efficacy of immune-modulators on resected mouse model will be more clinically relevant as resection itself enhances immune activation and may in fact be supportive for the immunotherapy being tested.

## Methods

**Cell lines and cell culture: mouse tumor cell lines**. CT2A and GL261 were cultured in Dulbecco's modified Eagle's medium (DMEM; GIBCO) supplemented with 10% fetal calf serum (Valley Biomedical Inc.) and 1% penicillin/streptomycin (Invitrogen). 005-GFP, Mut3 and Mut4 cells were cultured in 1:1 neurobasal medium (GIBCO) supplemented with 1% penicillin/streptomycin (Invitrogen), 1% N₂ supplement, 2 μg/ml heparin (sigma), B27(Invitrogen/ GIBCO), 20 ng/mL of human EGF (R&D Systems), and 20 ng/mL of human FGF-2 (fibroblast growth factor; PeproTech).

**Engineering tumor cells**. GFP-Firefly luciferase-puromycin was inserted into Retroviral vector MSCV and virus packaged along with pCL-Eco using lipofectamine 3000 (thermofisher scientific). Viral supernatant was used to infect CT2A, GL261, Mut3, and 005-GFP tumor cells. For 005 GFP tumor cells, cells were selected with 5 μg/ml puromycin. For CT2A, GL261, and Mut3, GFP-positive cells were sorted on BD Biosciences LSRFortessa. Fluc activity was confirmed by plating titrating cell numbers with 0.1 mg/ml luciferin on a luminometer at 0.5 s/well. These cell lines were used for all in vivo experiments.

**Establishing syngeneic mice tumors and in vivo imaging**. Six to eight-week-old C57BL/6 (Charles river laboratories) were anaesthetized and immobilized on a stereotactic frame. Totally, $1 \times 10^5$ cells for CT2A, GL261, and 005 GFP-Fluc lines and $5 \times 10^5$ Mut3 tumor cells in 4 μl phosphate-buffered saline (PBS) were implanted 2 mm deep, 2.5 mm lateral from bregma, and 2.5 mm ventral from dura in the right hemisphere. Mice were injected with luciferin i.p and imaged for Fluc activity once weekly. On reaching endpoint, mice were perfused, and brains were harvested for downstream processing. All in vivo procedures were approved by the BWH Institutional Animal Care and Use Committee.

**Mouse GBM surgical resection**. Cranial windows were created by removing a small portion of around 5 mm diameter of the skull in anesthetized C57Bl/6 mice. Seven days later, $2 \times 10^5$ CT2A-GFP Fluc cells were superficially implanted in the cranial window 2.5 mm lateral from the bregma and 0.7 mm deep. Tumor growth was checked by bioluminescent imaging every 3 days. For fluorescence-guided resection, anaesthetized mice were immobilized and superficial tumor was exposed under Leica surgical microscope. Fluorescent tumor was resected to reduce the tumor volume. Bleeding was controlled by applying pressure with cotton swab. The wound was copiously irrigated with PBS and the skin was closed by suturing. Pre- and post-resection tumor burdens were analyzed by bioluminescent imaging. Four days later, tumors were harvested.

**Western blotting**. Cells were washed with 1× PBS and lysed with lysis buffer (NP-40 (Sigma) supplemented with phosphatase (Sigma) and protease inhibitors (Roche) on ice and incubated for 15 min. For preparing tissue lysates, end-stage tumors were isolated and homogenized with lysis buffer. Cell/tissue lysates were clarified by centrifugation and supernatant was used to determine protein concentration. Totally, 10–30 μg of protein was resolved on sodium dodecyl sulfate polyacrylamide gel electrophoresis and transferred to a polyvinylidene difluoride membrane. P53 (CST), PTEN (CST), p-AKT (CST), AKT (CST), and vinculin antibodies were used followed by horseradish peroxidase (HRP)-conjugated secondary antibody for chemiluminescent detection.

**Mice brain tissue harvest**. Mice were extensively perfused with 30 ml of PBS by cardiac puncture. The fluid coming out of the mouse started to run clear and the changes in the color of the liver were suggestive of a good perfusion[61]. For mice harvested for IHC, they were further perfused with 20 ml of 4% paraformaldehyde (PFA). Fixation tremors were observed followed by body hardening that are a true indicator of successful perfusion. Brains were harvested and processed immediately for CyTOF, stored in RNAlater for RNA isolation or cryo-preserved for IHC.

**GBM patient samples**. The brain tumor samples were collected under 10–417, an institutional banking IRB approved protocol. The samples were distributed under tissue sub usage protocol approval. All patients undergoing a brain tumor surgery at the Brigham are open to this banking protocol at the time of surgery. The IRB is approved by the DF/HCC IRB and signed consent was obtained from all patients. Freshly isolated tumor tissue was harvested and immediately processed within an hour of surgery. Totally, 10 ml blood was separately collected from these patients at the time of surgery and was processed with the tumor tissue. In parallel, tissue sample was genotyped. All the 5 samples obtained were WHO grade IV GBM that were IDH1 WT with a mix of MGMT methylated and unmethylated status. Two patients had EGFR amplification, four had PTEN loss, three patients showed CDKNA2 loss, and one showed TP53 loss on genetic screening.

**Tissue processing and histochemistry**. Brains were harvested as described above. Brains were transferred to 30% sucrose from PFA and then sectioned on a cryostat. Totally, 8 μM sections were rehydrated in water and stained with hematoxylin (Richard Allan scientific) for 1 min. The stained slides were washed under tap water for 5 min followed by counter staining with Eosin Y (Sigma) for 30 s. Slides were gradually dehydrated in 95% ethanol (twice, for 1 min each) and then 100% ethanol (twice, for 1 min each), cleared with xylene and mounted with cytoseal XYL (thermo Scientific). Brain sections were immune-stained with antibodies against CD31, Ki67, CD3, CD4, CD8, Foxp3, and CD68 followed by alexa fluor 556 conjugated secondary antibody (Invitrogen) or HRP-conjugated secondary antibody (Vectastain ABC kit).

**RNA isolation and RNA sequencing**. Brains were harvested as described above. Tumor tissue was stored in RNA-later until all samples were collected. RNA was isolated from the tumor tissue using RNeasy mini kit (Qiagen) followed by on-column DNase treatment with RNase-free DNase set (Qiagen). The concentration for RNA samples were checked using Qubit RNA HS assay kit (ThermoFisher). Extracted RNA was plated at a concentration of 5 ng/μl and whole transcriptome sequencing was performed on Illumina at the Broad Technology Labs. Reads were aligned using STAR v2.4.2a with GENCODE mouse genome. Duplicate reads were marked and summary metrics were gathered using Picard tools and RNA-SeQC.

**Tissue processing for CyTOF**. Mouse tumor samples were processed as previously described[22]. Briefly, right half of the brain was separated and minced

with a scalpel on ice in calcium containing 1× HBSS (GIBCO) supplemented with 1 mg/ml Collagenase IV (Sigma) and 0.25 mg/ml DNase I (Sigma) and incubated for 1 h with intermittent shaking at 37 °C. Tumor Infiltrating immune cells (TIICs) were separated by a 30 and 70% percoll gradient centrifugation.

Patient tumor pieces were minced and suspended in 1 ml of RPMI without phenol red supplemented with 1 mM GlutaMAX (Life Technologies), antibiotic–antimycotic (Life Technologies), 2 mM MEM nonessential amino acids (Life Technologies), 10 mM HEPES (Life Technologies), $2.5 \times 10^{-5}$ M 2-mercaptoethanol (Sigma-Aldrich) and 200 µg/ml final Liberase TL enzyme mix (Roche). The tissue was digested for 10 min at room temperature followed by three low-speed (200 g) spins to separate the cells from debris and a total of 6.40E + 05-4.00E + 06 cells were present in each sample. The human tumor cells or mouse TIICs were resuspended in Cryostor CS10 (BioLife Solutions) for long-term cryopreservation in liquid nitrogen. Between 0.5 and $1.0 \times 10^6$ cells were used for each sample post thawing.

**CyTOF staining and data analysis**. Isolated TIICs were stained with 5 µM of cisplatin viability staining reagent (Fluidigm) for 5 min. Cisplatin-based viability reagent was titrated with known concentrations of dead cells. Cells that had been heat killed were spiked in with live autologous cells at a known concentration. The mean intensity threshold of cisplatin for dead cells was determined from this data. After centrifugation, Human/mouse TruStain FcX Fc receptor blocking reagent (BioLegend) was used at a 1:100 dilution in CSB (PBS with 2.5 g bovine serum albumin (Sigma Aldrich)) for 10 min followed by incubation with pretitrated metal-conjugated surface antibodies for 30 min. Cell lineage defining markers were selected to cover most of the immune populations. T-cell subsets can be identified by CD3, CD4, CD8, and TCRgd while NK cells can be defined by CD49b and NK1.1[62]. B cells express B220 and IgM is the most common isotype of B cells reported in tumors[63]. Various populations of innate immune cells including various monocyte/macrophage populations and dendritic cells can be identified by a combination of CD11b, CD11c, SiglecF, MHC-II, F4/80, CD103, CD115, and CD64[64,65]. Ly6C and Ly6G can help delineate various granulocytic populations[65] Apart from the composition of immune cells in the tumor micro-environment, the activation/inhibition status of these immune cells dictates the immunological response to the developing tumor. For functional characterization of T cells, we added Tim3, Lag3, KLRG1, PD-1H, CD25, CD44, CD62L, and CD86[66]. CD39 is an ectonuclease that serves as an inhibitory marker on cell surface of many different immune cell populations[67,68]. All antibodies were obtained from the Harvard Medical Area CyTOF Antibody Resource and Core (Boston, MA) or Fluidigm. To identify single cell events, DNA was labeled for 20 min with an 18.75 µM iridium intercalator solution (Fluidigm). Samples were subsequently washed and reconstituted in Milli-Q filtered distilled water in the presence of EQ Four Element Calibration beads (Fluidigm) at a concentration of $1 \times 10^6$ cells/mL and acquired on a Helios CyTOF Mass Cytometer (Fluidigm).

The raw FCS files were normalized to reduce signal deviation between samples over the course of multiday batch acquisitions, utilizing the bead standard normalization method[69]. To correct for any signal spillover-related issues, single color stained beads were run and used for compensation as previously described[70] (Supplementary Fig. 6B). These compensated files were then deconvoluted into individual sample files using a single-cell based de-barcoding algorithm[71] and compensated data was subjected to preliminary gating for doublet and dead-cell exclusion (Supplementary Fig. 6C). Supplementary Fig. 6D represents the analysis workflow down-stream of preliminary gating. Utilizing Cytobank, equal sampling was performed on CD45+ cells in all samples, followed by dimensionality reduction with the Barnes-Hut implementation of the t-SNE algorithm, i.e., viSNE[72]. This analysis presents the multidimensional data in a transformed two-dimensional space, while aiming to preserve its local and global structure. Downstream clustering was performed with the FlowSOM algorithm[73] on the dimensionality reduction channels generated by viSNE (i.e., tSNE-1 and tSNE-2). The main immune subsets were phenotypically isolated by choosing 7 metaclusters and entire sample cohort was run through an unsupervised-learning algorithm PhenoGraph[74] to estimate the optimal number of deeper phenotypic cluster. Utilizing an optimized version of k nearest neighbors clustering, PhenoGraph was able to determine the best-fit cluster number that partitions the data into discrete modules, allowing for detection of rare populations. Heat map values were generated by normalizing each marker to the minimum expression across all user defined clusters. Clusters were organized into groups in the heatmap by hierarchical average linkage clustering.

**Flow cytometry staining and data analysis**. TIICs were stained with 5 µM live/dead fixable violet dye (Thermofisher) followed antibody staining with Tim3-PE (clone-RMT3-23; biolegend; Catalog number-119703); CD4-BV711 (clone-RM4-5, biolegend, Catalog number-100549), CD8-PerCP (clone-53-6.7, biolegend, catalog number-100731) at pretitrated concentrations. Stained cells were fixed with 4% paraformaldehyde prior to running them on BD Fortessa. FCS files were analyzed on FlowJo (version 10.3.1).

**Statistical analysis**. Data were analyzed by Student $t$ test with Holm–Sidak multiple corrections applied on GraphPad prism. Data is expressed as mean ±

SEM. For Kaplan Meier plots, each pair of survival curves were compared using log-rank (Mantel–Cox) test. Differences were considered significant at adjusted $P \leq 0.05$ (*) and $P \leq 0.005$ (**), $P \leq 0.0005$ (***).

**Reporting summary**. Further information on research design is available in the Nature Research Reporting Summary linked to this article.

## Data availability

RNA sequencing data is publicly available at https://www.ncbi.nlm.nih.gov/geo/query/acc.cgi?acc=GSE151414 and CyTOF data has been uploaded at ImmPort [https://www.immport.org/shared/home] with accession number SDY1637. All other relevant data are available in the article, Supplementary Information, or from the corresponding author upon reasonable request.

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

## Acknowledgements

The authors would like to acknowledge Drs. Martuza and Rabkin for providing 005 tumor line and Dr. Wakimoto for critical reading of the paper. This work was supported by RO1 CA138922 (K.S.), R01 CA204720 (K.S.), and CA140744 (K.S.). The authors would like to thank Lorena Pantano and John Hutchinson of the Harvard Chan Bioinformatics Core, Harvard T.H. Chan School of Public Health, Boston, MA for assistance with Bioinformatic analysis of RNA seq data. We would like to thank Michael Wheeler and Dr. Francisco J. Quintana for their help with RNA sequencing sample preparation. We would also like to thank Susan R. Paul and Patrick Reeves for their help with CyTOF panel design and Dr. Steven Piantadosi for his valuable inputs in statistical analysis.

## Author contributions

J.K.K.: conception and design, provision of study material, collection and assembly of data, data analysis and interpretation, paper writing, and final approval of the paper; N. C.: collection and assembly of data, data analysis and interpretation, and final approval of the paper; J.Ke.: provision of study material, collection and assembly of data, data analysis and interpretation and final approval of the paper; A.C.: collection and assembly of data, and final approval of the paper; J.D.: collection and assembly of data, and final approval of the paper; L.W.B.: collection and assembly of data, and final approval of the paper; J.L.: data analysis and interpretation and final approval of the paper; K.S.: conception and design, provision of study material, data analysis and interpretation, paper writing, and final approval of the paper.

## Competing interests
