## [Peer Review File · Nature Communications]

Reviewers' comments:

Reviewer #1 (Remarks to the Author): (expert in CyTOF and brain immune system)

In this study, Khalsa et al generated syngeneic GBM31 tumor models and analyzed the resulting immune profiles using high dimensional tools including RNA-sequencing and CyTOF as well as correlative immunohistochemistry. They show that immune response in the tumors is characterized by immune-suppressive phenotype. They then show that tumor-resection increases CD8+ T cells and monocyte/macrophages while decreasing Ly6G+ myeloid cells. They then validate some of these findings in tumors from GMB patients.

This study is potentially of importance and can be relevant for our understanding of GMB, however, some crucial aspects are still limited.

1. Fig. 2: The identification of microglia cells as CD45 low / CX3CR1+ is not sufficient. The authors should add at least CD11b as an additional defining marker for this cell population.

Additionally, in an inflammatory, the depiction of microglia cannot solely rely on conventional markers; the use of additional markers (such as Tmem119, CD49d) is required to properly segregate between microglia and blood-derived myeloid cells.

2. Throughout the manuscript, the statistical description in the figure legends and in the material and methods section is lacking.

Please add the number of experiments performed, number of repeats within each experiment, what each repeat represents (mouse/cell/pooled sample), exact p values, etc. It will be more informative if the authors change the plots to individual dot plots and not bar charts. In many plots, the statistical test is not indicated (no indication for significance or insignificance). It seems the authors draw several conclusions according to trend-based data (Figure 4,5,6).

3. Fig. 4 and 5: did the authors apply here any multiple corrections for the multiple t-tests?

4. Fig. 5: The are important data that are missing in the experimental description of the resection experiment. Was the tumor monitored during this experiment?

The surgical procedure on its own can trigger an immunological response; the authors should add an additional control group for the surgical resection without any tumor cells.

5. How did the authors determine the "end-stage"? this is somewhat vague.

Were the tumors/brains extracted on the same day? size? upon a humane endpoint?

According to Figure S1, the various tumor cell lines have different growth rates. This is likely to be an important parameter when determining the extraction time of the tumor/brain, which in turn, may affect the immunological parameters of the system. The authors claim differences between tumor cell types, is it possible that these differences are the consequence of the selected "end-stage".

6. It would be interesting to see the differences between the tumor-related effects in the blood compared to those of the brain. Perhaps the immunological signature of the blood can correlate the underlying processes of the brain (at least refer to the known literature when possible).

7. Did the authors validate that their perfusion was successful (may be crucial for the analysis)?

8. Fig.6: The immune populations that are in part A of the figure do not match those that are in part B (APCs/Macrophages), why?

The conclusion here appears to be based on a small number of parameters and limited statistics. This information should be added (how many patient samples were used here? what were their parameters? how were they extracted?) and the limitations acknowledged.

The figure legend is missing some of the cell type markers.

The B cells definition should be revisited. Please indicate also the IgG+ B cells as the authors indicate only IgM as the entire B cell population.
Please refer also to the microglia population. Despite the variability between the samples, these cells can still be analyzed and may have some unique phenotypes.
Overall the data in this figure appears to be limited in spite of the use of a high-dimensional tool.

9. Figure S6: Please add the exact expression level for each marker in SPADE.

B cells were defined with only the IgM+ subset.

CD64 is also expressed by microglial cells.

What is the marker for each SPADE plot?

The definition of microglial cells is similarly lacking here (CD11b for example).

10. Figure S7: what are the expression levels for each viSNE plot? In addition to min-max.

11. Are there any flow cytometry validations for the CyTOF data?

12. What technology was used in Figure S1 to track tumor size? Only fluorescence microscopy analysis? Was there any in vivo tracking (IVIS)?

13. Fig. 2E: the microglia appear to be CX3CR1 negative cells? In addition, CD45 changes and thus is very limited as a sole marker for microglia under inflammation.

14. Fig. S3: What are "Residual", "center" and "Beads-ve"? How was the Cisplatin gate determined?

15. Fig. S7: it is not clear what was the sample

16. Fig. S9, Fig. S10: What is the method used in these figures? How were the heat-maps calculated (min-max does not necessarily mean the expression level is relevant)?

17. Fig. 1, Fig.3: Please define what is the total "ungated" events/cells.
a more detailed description of the process is needed (perfusion volume, the used media)

Reviewer #2 (Remarks to the Author): (expert in glioblastoma and tumor microenvironment)

Overall this is an exciting and timely manuscript that uses various transplantable murine cell line-based models to compare to human GBM with respect to TIICs. Authors use high dimensional technologies such as CyTOF and scRNA-seq complemented with basic IHC to look for similarities and differences between human and murine tumors and to identify the best mouse model that closely clusters with human GBM TIIC profile. The manuscript focuses on a significant problem, which is the choice of the right models for GBM immune studies. But it also comes short with the choice of only the cell-line based transplantable models, to compare to human GBM. If the goal was to identify the better murine models, the broader range of models should have been included.

Besides, several significant weaknesses need to be addressed before publication; as it is it gives the impression that the manuscript was a bit rushed into publication without careful consideration of some essential details that can change the interpretation of results. Cell populations are defined based on some markers that are not always accurate and matching what is known in the literature. Some critical details are missing, which is essential to back up some of the claims.

Major comments

The material and method section is completely missing information on human GBMs used in the comparison. It is the last paragraph and the most important with minimum details on sample size and types of the tumors that are used.

005 line that authors find mimicking human glioblastoma, unfortunately, does not represent the genetics of human disease. H-Ras is the driver mutation, how many glioblastoma patients have H-Ras mutation? Less than 1% based on some and close to zero based on the other publications. 005 line should have constitutively active Akt based on the author's text (page 3, line 77) and reference 15th. Cell line 005 was derived from tumors that are generated with constitutively active Akt co-injection, while western blot shows no pAKT and high levels of PTEN opposite of what one would expect to see based on publication and genotype they refer Figure 1B. This should be clarified.

Based only on Figure 6 I am not very convinced that 005 is any better than CT2A, authors should comment on this

Tumors that are called Mut3 are derived from the Mut3 line, which is 129Svj/C57Bl6 hybrid background. The authors transplanted them into the C57Bl6 strain; they should clarify whether they acquired the Mut3 line from pure C57Bl6

There are no survival curves included in the manuscript (tumor-bearing mice transplanted with various tumor cell-lines); it is tough to estimate when do tumors develop when various lines are injected. Some of the lines are cultured in serum others in serum-free conditions; the number of cells injected into mice also vary depending from line to line. These are significant variables that should be considered.

These lines have been cultured in vitro for a long time that resulted in more aggressive cells that have been transplanted into the mice in excessive numbers. If the tumor cell growth rate is too fast there is a possibility that it may result in a dramatic difference in the recruitment of TIICs into the microenvironment. May this explain the fact that from all the tumors only 005 that shows similarity to humans and it has the slowest growth and is poorly vascularized, authors should comment on it?

Although authors refer to the resection of tumors, which is a partial resection, there are no details provided about the procedure of partial resection in the material and method section. How did they stop the excessive bleeding due to resection? What part of the tumor was resected? How did the author ensure similar resection from mouse to mouse? When comparing resected tumor profile to non-resected, how did they normalize four days difference that growth can cause? The resection creates an immediate inflammatory reaction and bleeding? In addition, only part of the tumor resected region when comparing to whole unresected tumors it is not the same.

Although Authors state that they did not focus on the total number of TIICs in various models but instead concentrated in their composition (line 192-194), in my opinion, it is critical to look at absolute numbers, especially when comparing human versus mouse

Figure 2A I am not sure I understand what is CD11c+microglia, CD115+microglia, CD11b+ positive microglia- none of these markers are specific only for microglia? It will be helpful to clarify.

MDSC/neutrophil cluster – not sure what it means. MDSC is a functional cell that can suppress the function of T-cells, can be either monocytic or neutrophilic. Authors do not have any functional assay in the manuscript, so they should be careful based on single-cell or CyTOF defining the functional myeloid cell. The most these technologies provide is a possible presence of clusters, that should be called and interpreted carefully.

Figure 2 A is extremely challenging to follow the gating strategy, and the FACS plots, especially the gating strategy for tumor-associated microglia. Total % of microglia in tumors is about 6.5%? Do authors see a difference in tumor-associated microglia in various murine tumor models? The % microglia seems low can it be partially attributed to the aggressive growth of malignant tumor cells and discrepancy that creates with the timing of tumor growth and microglia infiltration. In human tumors %, microglia are documented to be much higher. In the naïve CD11b/CD45 staining, there is a large population of CD45+CD11b- negative cells, seem very high for a naïve brain? The authors should clarify. Basic multicolor FACS for myeloid and T-cell panels if included can be very helpful.

Interestingly, authors do not see a difference in CD68+ macrophages in four different tumor types that have different origins and mutational profile. This is in contrast to many published works in the glioblastoma field, demonstrating that NF1 loss in human and murine models results in increased macrophage infiltration and correlated with high macrophage gene signature both in genetically engineered mouse models and MES GBM subtype. Authors should comment and discuss their findings and talk about differences with published literature.

Figure S6 has very low-quality IHC images and would benefit from sharper and clear staining. CD68 staining is inferior quality, and I am not sure how authors quantified the number of cells based on surface marker. I would advise to stain for IBA1 and quantify % positive area.

Ly6C+ macrophages were classified as mononuclear MDSCs? Ly6C+ labels monocytes and, it is not a marker for macrophages. Newly infiltrated monocytes from circulation downregulate Ly6C upon differentiation into tissue macrophages. The authors should be cautious with terminology. And again, MDSC is a function of a cell and is not defined by Ly6C+CD11b+, as authors refer in the abstract. Without functional assay to show that Ly6C+CD11b+ cells isolated from tumors can suppress the function of T-cells, they should be called monocytes.

In the material and methods section, it states that Student's t-test was used, but there are instances that more than one group of comparison and that is not appropriate to use - Sfigure 4

Figure 6 B is the 005 missing the error bars, or they are just that consistent? How many samples are included per tumor type?

Reviewer #3 (Remarks to the Author): (expert in glioblastoma and RNA-seq)

Khalsa et al used a combination of H&E, IHC, RNAseq and CyTOF to perform comprehensive immune profiling on syngeneic mouse models of GBM. For each model, they additionally use CyTOF to compare the immune profiles between surgically resected and non-resected tumors and compare these results to those obtained from surgical resections from GBM patients. The findings represent an advance over what is currently known about how the immune response in syngeneic mouse models of GBM compares to that in human GBM, and the data presented represent an excellent resource for future studies. The major concern of this manuscript relates to a lack of statistical testing that is persistent throughout the results.

Major comments.

1. The statistical information throughout the manuscript is lacking, and related, the number of samples analyzed in the many comparisons is missing. Both are essential issues that need to be addressed. A non-comprehensive list of examples:

- Figure 1b, comparison of CD31 staining,

- 'showed dramatically lower SiglecF+ macrophage (CD11b+F4/80+SiglecF+MHCII+) and eosinophil (CD11b+SiglecF+MHCII lo) populations'
 - 'CT2A tumor samples had more than two-fold higher monocytes/macrophages 215 compared to all the other tumor types'
 - The frequency differences outlined in Fig. 3B
 - The exhaustion marker analyses for T cells outlined in Supplementary Figure 6C. If the T cells do not significantly differ in their expression of exhaustion markers across models, this should be noted as it is valuable information for future experiments that use these models to study the effect of checkpoint inhibitors.
 - There are no p-values for the analyses outlined in Supplementary Figure S8, despite claims that the number of T cells differs between subgroups of the models.
 - There are no p-values for the CD8+ T cell-specific differences outlined in Figure 4B or Supplementary Figure S9.
 - There are no p-values for any of the immune cell differences outlined in Figure 5.
 - There is no correlation coefficient demonstrating the similarity in immune profiles between patient GBMs and syngeneic mouse models.
2. Are the differences in immune-cell related gene expression levels between specimens from the same model comparable, shown in Fig 1E, lesser or greater than the inter-model variability? For example, the differences between Mut3 and control seem negligible. Again, statistical testing is needed.
 3. CT2A (and Mut3) tumor samples were classified as immunologically inert, GL261 (and 005) tumors as immune-active tumors. How do those observations compare with the better efficacy of CT2A tumors compared to GL261 when treated with autologous tumor lysates, reported in Genoud et al., *Oncoimmunology* 2018 (PMID 30524896).
 4. Immunologically inert tumors are suggested to contain fewer infiltrating immune cells, and this result seems as odds with the many reports that the frequency of innate myeloid immune cells is higher in mesenchymal glioblastomas, a category of GBM that is associated with worse outcomes compared to non-mesenchymal glioblastomas, (Wang et al, *Cancer Cell* 2017, PMID 28697342 and Engler, *PLOS One*, 2012, PMID 22937035).
 5. Tumors were compared four days post resection; clarification is needed what this residual material represents since presumably there are few tumor cells left post-resection. Was any histology/immunohistochemistry performed to demonstrate presence of tumor cells post-resection? And what is the statistical significance of the findings?
 6. Specimens from how many GBM patients were analyzed? Since GBMs from different expression subtypes have been shown to have very different immunological profiles (Wang et al, *Cancer Cell* 2017, PMID 28697342), the expression subtype of these GBMs needs to be reported to put findings into context.
 7. The data outlined in the manuscript together represent an excellent resource for biologists to probe when planning experiments and need to be made accessible for exploration.

Minor comments:

1. Figure 1C: It would be helpful to have a box that indicates where in the 4X images the 20X images were taken.
2. The manuscript would benefit from a brief discussion of why each marker was chosen. What cell types were the author's hoping to find? What specific phenotypes were the authors searching for? Why were these markers the best markers for comparing each syngeneic model?
3. The asterisks indicating significance should be outlined in each figure legend rather than in the methods section.
4. What is the difference between a classical CD8+ T cell (which the authors also refer to as functional) and a tumor-reactive CD8+ T cell?

Response to Reviewers' comments:

Reviewer #1

In this study, Khalsa et al generated syngeneic GBM31 tumor models and analyzed the resulting immune profiles using high dimensional tools including RNA-sequencing and CyTOF as well as correlative immunohistochemistry. They show that immune response in the tumors is characterized by immune-suppressive phenotype. They then show that tumor-resection increases CD8+ T cells and monocyte/macrophages while decreasing Ly6G+ myeloid cells. They then validate some of these findings in tumors from GMB patients. This study is potentially of importance and can be relevant for our understanding of GMB, however, some crucial aspects are still limited.

Response: We would like to thank the reviewer for her/his appreciation of our study and constructive comments.

1. Fig. 2: The identification of microglia cells as CD45 low / CX3CR1+ is not sufficient. The authors should add at least CD11b as an additional defining marker for this cell population. Additionally, in an inflammatory, the depiction of microglia cannot solely rely on conventional markers; the use of additional markers (such as Tmem119, CD49d) is required to properly segregate between microglia and blood-derived myeloid cells.

Response: As suggested by the reviewer, we have identified microglial cells as CD45low/CX3CR1+ /CD11b medium in Fig 2B. Previous studies have shown that microglial cells upregulate MHCII under inflammatory conditions such as the tumor micro-environment, (Schetters et al, 2018). We have observed upregulation of MHCII in tumor bearing mice as shown in in Fig. 2E.

2. Throughout the manuscript, the statistical description in the figure legends and in the material and methods section is lacking. Please add the number of experiments performed, number of repeats within each experiment, what each repeat represents (mouse/cell/pooled sample), exact p values, etc. It will be more informative if the authors change the plots to individual dot plots and not bar charts. In many plots, the statistical test is not indicated (no indication for significance or insignificance). It seems the authors draw several conclusions according to trend-based data (Figure 4,5,6).

Response: We have now added the statistical description in the "figure legends" and "material and methods" section. We have also included details on number of samples and number of repeats within each experiment in the figure legends and changed data representation of all experiments to dot plots. We have performed Student's T test with Holm-Sidak corrections applied for multiple t-tests and statistical significance was reported with * for $p < 0.05$ and ** for $p < 0.005$ and is stated in each figure legend.

3. Fig. 4 and 5: did the authors apply here any multiple corrections for the multiple t-tests?

Response: We have applied Holm-Sidak corrections for the multiple t-tests through-out the manuscript now.

4. Fig. 5: The are important data that are missing in the experimental description of the resection experiment. Was the tumor monitored during this experiment?

The surgical procedure on its own can trigger an immunological response; the authors should add an additional control group for the surgical resection without any tumor cells.

Response: Tumor cells were engineered to express firefly luciferase and tumor growth was monitored by bioluminescence imaging, as shown in Fig. S1B and Fig. S1C. We agree with the reviewer that surgical procedure can result in an immunological response and we have addressed that in our previous publication (Choi et al, 2017). Specifically, we observed a modest yet significant increase In T cell and dendritic cell

frequency until day 4 in surgical resection in the brains of naïve mice. However, the increase in T cells and dendritic cells was much higher where tumors were surgically resected.

5. How did the authors determine the "end-stage"? this is somewhat vague. Were the tumors/brains extracted on the same day? size? upon a humane endpoint? According to Figure S1, the various tumor cell lines have different growth rates. This is likely to be an important parameter when determining the extraction time of the tumor/brain, which in turn, may affect the immunological parameters of the system. The authors claim differences between tumor cell types, is it possible that these differences are the consequence of the selected "end-stage".

Response: We agree with the reviewers that stage of the tumor can have implications on immunological response. Therefore, in this study, only end-stage tumors for all tumor types were compared. The end-stage of the tumor was defined for each tumor type as an in-vivo bioluminescent photons/min of 10^7 where neurological symptoms develop in mice due to increased intra-cranial pressure. TILs were isolated, frozen down in liquid nitrogen and thawed together for CyTOF staining. As different tumor types grow at varying rates, tumor cell lines were implanted at staggered time points.

6. It would be interesting to see the differences between the tumor-related effects in the blood compared to those of the brain. Perhaps the immunological signature of the blood can correlate the underlying processes of the brain (at least refer to the known literature when possible).

Response: We agree with the reviewers and we performed comparative blood and TIL immune phenotyping on matched tumor and blood obtained from GBM patients, as shown in supplementary Fig. S14. Our new data indicates that tumors have drastically different immune cell profile as compared to blood from patients or healthy donors. Of the total CD3+ cells, healthy donor blood has 72%/3% CD4 T cells/Tregs while blood from patients has 48%/7% conv. CD4/Treg cells. On the contrary, tumor infiltrating T cells have 16% conventional CD4 T cells and 13% Tregs. This data is in concordance with the previous studies from Mohme et al, 2018. Furthermore, T cell subset analysis shows total CD8 T cells are more in the tumor and CD103+CD8 T cells are exclusively present in the tumor.

7. Did the authors validate that their perfusion was successful (may be crucial for the analysis)?

Response: Yes. mice were extensively perfused with 30ml of PBS by cardiac puncture before harvesting tumor-bearing brain from mice. Specifically, the fluid coming out of the mouse started to run clear and the changes in the color of the liver were suggestive of a good perfusion (Gage et al, 2012). For mice harvested for immunohistochemistry, mice were further perfused with 20 ml of 4% paraformaldehyde. Fixation tremors were observed followed by body hardening that are a true indicator of successful perfusion. These details have been added to materials and methods.

8. Fig.6: The immune populations that are in part A of the figure do not match those that are in part B (APCs/Macrophages), why? The conclusion here appears to be based on a small number of parameters and limited statistics. This information should be added (how many patient samples were used here? what were their parameters? how were they extracted?) and the limitations acknowledged.

The figure legend is missing some of the cell type markers. The B cells definition should be revisited. Please indicate also the IgG+ B cells as the authors indicate only IgM as the entire B cell population. Please refer also to the microglia population. Despite the variability between the samples, these cells can still be analyzed and may have some unique phenotypes. Overall the data in this figure appears to be limited in spite of the use of a high-dimensional tool.

Response: We have now modified Fig. 5 (previously Fig. 6) which now includes an extensively characterized APC/Macrophage subset . Specifically, we performed the experiment again and added more markers in order to define immune cell populations as listed in Fig. 5 legend. Tissues from 5 WHO grade-IV glioma patients that were all IDH wildtype were stained for this study. The genotyping of the resected

material was done in parallel to the immune cell isolation and during the time of revising the manuscript, all the patient samples we obtained were IDH wildtype. As comparison of immune phenotypes of mouse models with patient samples from other GBM subtypes might yield different results, we have now included this in the discussion section of the manuscript.

We have also added the details of sample processing in the Material and Methods section of the manuscript.

Based on reviewers suggestion, the following changes were also made in the manuscript: 1) Cell type markers have been added in figure legends for Figure 3,4 and 5; 2) Most/all of B cells were all IgM+ in this study and figure legend in figure 3,4 and 5 has been modified; 3) Human microglia population has been identified as CD11b low, CD45 low, CD44 low and further segregated into resting and activated microglia based on MHC-II expression. These details have been added to figure legend of Fig. 5.

9. Figure S6: Please add the exact expression level for each marker in SPADE. B cells were defined with only the IgM+ subset. CD64 is also expressed by microglial cells. What is the marker for each SPADE plot? The definition of microglial cells is similarly lacking here (CD11b for example).

Response: We have now modified Supplementary Fig. S6 and marker information for each SPADE plot and expression levels were added. The reviewer has correctly pointed out that CD64 is a broader marker and therefore, we have renamed the population as Monocytes/Macrophage. Microglial cells were defined as CX3CR1+/CD45 low in the biaxial plot as previously reported in Korin et al 2017 study. Microglial cell (CX3CR1+/CD45 low) population had low CD11b expression as indicated in the SPADE plot and that cluster is distinct from the CD64+ monocyte/macrophage cluster.

10. Figure S7: what are the expression levels for each viSNE plot? In addition to min-max.

Response: We have now included a scale for each viSNE plot in Supplementary Fig. S7.

11. Are there any flow cytometry validations for the CyTOF data?

Response: Flow cytometry validation for CyTOF data has been previously reported by Gadalla et al, 2019. We have also verified CD4/CD8 frequencies and Tim3 expression on these populations as shown in supplementary Fig S3.

12. What technology was used in Figure S1 to track tumor size? Only fluorescence microscopy analysis? Was there any in vivo tracking (IVIS)?

Response: All tumor lines were transduced to express bimodal bioluminescent marker Firefly luciferase fused to GFP. Tumor burden was detected by luciferase imaging post injection of substrate D-luciferin intraperitoneally and plotted over time as shown in Fig. S1B and Fig. S1C.

13. Fig. 2E: the microglia appear to be CX3CR1 negative cells? In addition, CD45 changes and thus is very limited as a sole marker for microglia under inflammation.

Response: Microglial cells were first identified as CD11b^{low}/CD45^{low} and gating on these cells identified resting and activated microglia on the basis of CX3CR1 and MHC-II expression. In inflammatory conditions, microglia are known to upregulate MHC-II expression (Schettler, et al, 2018). Therefore, CX3CR1^{high}/MHCII^{low} were labelled as resting microglia and CX3CR1^{high}/MHCII^{high} as activated microglia. As can be seen in Fig. 2E, most CD11b^{low}/CD45^{low} cells are CX3CR1 positive.

14. Fig. S3: What are "Residual", "center" and "Beads-ve"? How was the Cisplatin gate determined?

Response: Post acquisition of a CyTOF sample, we generated an FCS file and the acquisition software determined the gaussian distribution of total metal counts across a given sample (i.e. the aggregate metal

detected from antibody, viability, and DNA based reagents). The gaussian parameters include: Center, Offset, Residual, and Width. These parameters are crucial for the elimination of outliers, e.g. doublets and debris. We also utilized a bead-based reagent for normalization of the CyTOF signal over time. After the normalization process was complete, the bead standards were removed to perform a bead specific exclusion gate, thus allowing us to retain our cellular events.

We have titrated our cisplatin-based viability reagent with known concentrations of dead cells. Cells that had been heat killed were spiked in with live autologous cells at a known concentration. The mean intensity threshold of cisplatin for dead cells was determined from this data. These details are now included in the Methods section of the manuscript.

15. Fig. S7: it is not clear what was the sample

Response: The sample used to obtain data shown in Fig. S7 was from GL261 tumors. We have now mentioned this in the figure legend.

16. Fig. S9, Fig. S10: What is the method used in these figures? How were the heat-maps calculated (min-max does not necessarily mean the expression level is relevant)?

Response: Heat map values (as shown in Fig. 2C) were generated by normalizing each marker to the minimum expression across all user defined clusters. We have now included this information in the Materials and Methods section of the manuscript. We have now removed data from Fig S9 and Fig S10 and replaced it with bar graph showing normalized mean expression levels of activation/inhibition markers as shown in Figure 4E.

17. Fig. 1, Fig.3: Please define what is the total "ungated" events/cells. a more detailed description of the process is needed (perfusion volume, the used media)

Response: The ungated events, as defined in Fig. 2,3,4 and 5 are live gated CD45 positive cells as shown in Supplementary Fig. S4C. We have now included a separate section on mouse brain harvesting in the Methods section of the manuscript. Mice were extensively perfused with 30ml of PBS by cardiac puncture. A clear fluid coming out of the heart and the color of the liver changes post-perfusion was suggestive of a good perfusion (Gage et al, 2012). For mice harvested for immunohistochemistry, mice were further perfused with 20 ml of 4% paraformaldehyde. Fixation tremors were observed followed by body hardening that are a true indicator of successful perfusion. Brains were harvested and processed immediately for CyTOF, stored in RNAlater for RNA isolation or cryo-preserved for IHC.

Reviewer #2

Overall this is an exciting and timely manuscript that uses various transplantable murine cell line-based models to compare to human GBM with respect to TIICs. Authors use high dimensional technologies such as CyTOF and scRNA-seq complemented with basic IHC to look for similarities and differences between human and murine tumors and to identify the best mouse model that closely clusters with human GBM TIIC profile. The manuscript focuses on a significant problem, which is the choice of the right models for GBM immune studies. But it also comes short with the choice of only the cell-line based transplantable models, to compare to human GBM. If the goal was to identify the better murine models, the broader range of models should have been included.

Besides, several significant weaknesses need to be addressed before publication; as it is it gives the impression that the manuscript was a bit rushed into publication without careful consideration of some essential details that can change the interpretation of results. Cell populations are defined based on some markers that are not always accurate and matching what is known in the literature. Some critical details are missing, which is essential to back up some of the claims.

Response: We would like to thank the reviewer for finding our study exciting & timely and for his/her constructive comments.

Major comments

The material and method section is completely missing information on human GBMs used in the comparison. It is the last paragraph and the most important with minimum details on sample size and types of the tumors that are used.

Response: We would like to thank the reviewer for pointing this out. We have now included the materials and methods section on human GBM studies. We have also included the details on sample size and types of tumors in the methods section. Specifically, tissues from 5 WHO grade-IV glioma patients that were all IDH wildtype patients were stained for this study and a total of $6.40E+05-4.00E+06$ cells were present in each sample.

005 line that authors find mimicking human glioblastoma, unfortunately, does not represent the genetics of human disease. H-Ras is the driver mutation, how many glioblastoma patients have H-Ras mutation? Less than 1% based on some and close to zero based on the other publications. 005 line should have constitutively active Akt based on the author's text (page 3, line 77) and reference 15th. Cell line 005 was derived from tumors that are generated with constitutively active Akt co-injection, while western blot shows no pAKT and high levels of PTEN opposite of what one would expect to see based on publication and genotype they refer Figure 1B. This should be clarified.

Response: We do agree with the reviewer that none of the mouse models of GBM truly recapitulate the human GBM genetically. However, our study attempts at revealing the similarities and differences in the immune cell make up in the tumor micro-environment of the commonly used mouse GBM models. The reviewer is correct that in GBM H-Ras mutation is very rare. However, NF1 mutation is frequent and it is possible that H-Ras mutation could mimic the inactivation of NF1 since Ras is downstream of NF1. 005 line was modified to have a constitutively active AKT by addition of a myristoylation signal to its N-terminus, resulting in translocation of Akt to the membrane. Higher exposure of pAKT blot shows low levels of the protein as shown in Figure A at the bottom of the response document. The main target of PTEN, a phosphatase, in the AKT pathway is to dephosphorylate phosphatidylinositol phosphate (PIP3). PIP3 leads to AKT recruitment to the membrane. Myristoylated AKT is already anchored to the membrane leaving phosphorylated status of PIP3, and thus, role of PTEN redundant in this modified AKT pathway.

Based only on Figure 6 I am not very convinced that 005 is any better than CT2A, authors should comment on this

Response: We have now expanded on the immune cell populations compared between mouse and humans as shown in Fig. 5 (previously Fig. 6). We also performed Spearman's rank correlation coefficient between GBM samples and each of the mouse models. R value for 005 and GBM patients was the largest as compared to other mouse tumor models.

Tumors that are called Mut3 are derived from the Mut3 line, which is 129Svj/C57Bl6 hybrid background. The authors transplanted them into the C57Bl6 strain; they should clarify whether they acquired the Mut3 line from pure C57Bl6

Response: Mut 3 cell lines obtained from 129Svj/C57Bl6 mice were passaged in C57Bl6 mice before utilizing them for this study.

There are no survival curves included in the manuscript (tumor-bearing mice transplanted with various tumor cell-lines); it is tough to estimate when do tumors develop when various lines are injected. Some of the lines are cultured in serum others in serum-free conditions; the number of cells injected into mice also vary depending from line to line. These are significant variables that should be considered.

Response: We have now included survival curves for the 4 mouse models represented in Fig. 1C. We agree with the reviewer that tumor cells have different culture conditions and number of cells injected into mice also vary depending on the use of the cell line. In this study, we made sure that injected tumor cell numbers and day of harvest post implantation were standardized, and all tissues were harvested at end-stage.

These lines have been cultured in vitro for a long time that resulted in more aggressive cells that have been transplanted into the mice in excessive numbers. If the tumor cell growth rate is too fast there is a possibility that it may result in a dramatic difference in the recruitment of TIICs into the microenvironment. May this explain the fact that from all the tumors only 005 that shows similarity to humans and it has the slowest growth and is poorly vascularized, authors should comment on it?

Response: The reviewer has raised a very valid point. In our studies, Mut3 is the slowest growing tumor model and takes the longest to reach end-stage, however it does not match the patient sample as well based on Spearman's correlation co-efficient shown in Fig. 5C. We have now included a description on this in the discussion section of the manuscript

Although authors refer to the resection of tumors, which is a partial resection, there are no details provided about the procedure of partial resection in the material and method section. How did they stop the excessive bleeding due to resection? What part of the tumor was resected? How did the author ensure similar resection from mouse to mouse? When comparing resected tumor profile to non-resected, how did they normalize four days difference that growth can cause? The resection creates an immediate inflammatory reaction and bleeding? In addition, only part of the tumor resected region when comparing to whole unresected tumors it is not the same.

Response: We have now added details on the resection procedure in the material and methods section. Specifically, tumors were partially resected by fluorescence guided microscopy. We performed careful microscopic resection up to the tumor-tissue interface while leaving margins of the dura intact. There was limited bleeding in mice due to the procedure that could be controlled by applying pressure on the resection site with cotton swab. By this method, much of the core of the tumor was removed. Tumor burdens were reduced from $5.1 \times 10^5 \pm 79005$ by a log fold to $2.1 \times 10^4 \pm 2896$ (average \pm SE) as can be seen in Fig. 4B. There was little variability between mouse to mouse in post-resection tumor burden as can be appreciated by a small error bar post resection. The reviewers are correct at pointing out that resection results in bleeding that results in immediate inflammation. We have addressed this in our previous publication (Choi et al 2017) where we have compared naïve (no resection), tumor-free resected mice, unresected tumor-bearing mice and resected tumor-bearing mice and observed a modest, yet significant increase in T cell and dendritic cell frequency until day 4 in surgical resection without any tumor cells. However, the increase in T cells and dendritic cells was much higher where tumors were surgically resected. Tumor burden is different in resected versus unresected tumors as shown in Fig. 4B and the question we are trying to address is the implication of removal of the inhibitory tumor core on immune response. For comparing equivalent brain portions, tumors were implanted in the right hemisphere of the brain and we harvested right side of the brain for resected and unresected tumor-bearing mice for data in Fig. 4B.

Although Authors state that they did not focus on the total number of TIICs in various models but instead concentrated in their composition (line 192-194), in my opinion, it is critical to look at absolute numbers, especially when comparing human versus mouse

Response: We agree with the reviewer that total number of TIICs would be an interesting and informative readout. However, the tumor tissue size from patients is not consistent and therefore there is high variability in the number of tumor cells isolated making comparisons difficult.

Figure 2A I am not sure I understand what is CD11c+microglia, CD115+microglia, CD11b+ positive microglia- none of these markers are specific only for microglia? It will be helpful to clarify.

Response: There are 3 metaclusters that have low CD45 expression (Metacluster 4,5 and 6) that could be potentially microglial subsets. All of these have high CX3CR1 expression (Wolf,Y, et al 2013). Metacluster 4 has high CD11c and CX3CR1, defined as CD11c+ microglia. Metacluster 5 has CD115 expression as well, defined as CD115+ microglia. Ly6C expression can be seen in Metacluster 6, however with low CD45 expression and thus, we defined this as Ly6C+ microglia/monocytes. This description has now been included in the result section of the manuscript.

MDSC/neutrophil cluster – not sure what it means. MDSC is a functional cell that can suppress the function of T-cells, can be either monocytic or neutrophilic. Authors do not have any functional assay in the manuscript, so they should be careful based on single-cell or CyTOF defining the functional myeloid cell. The most these technologies provide is a possible presence of clusters, that should be called and interpreted carefully.

Response: We agree with the reviewer and have renamed this cluster to be Ly6G+ myeloid cells/neutrophils.

Figure 2 A is extremely challenging to follow the gating strategy, and the FACS plots, especially the gating strategy for tumor-associated microglia. Total % of microglia in tumors is about 6.5%? Do authors see a difference in tumor-associated microglia in various murine tumor models? The % microglia seems low can it be partially attributed to the aggressive growth of malignant tumor cells and discrepancy that creates with the timing of tumor growth and microglia infiltration. In human tumors %, microglia are documented to be much higher. In the naïve CD11b/CD45 staining, there is a large population of CD45+CD11b- negative cells, seem very high for a naïve brain? The authors should clarify. Basic multicolor FACS for myeloid and T-cell panels if included can be very helpful.

Response: We have re-plotted Fig. 2E. Cells were pre-gated for live cells as shown in Supplementary Fig. S4C. The tumor-bearing panel of Fig. 2E that was present in the initially submitted manuscript was an outlier, We have now included bi-axial plot from a mouse that is more representative of majority of tumor-bearing mice used in the study. First, we have gated on CD11b low/CD45 low cells as microglia. On these, CX3CR1+ve, MHCII low cells were gated as resting microglial cells and majority of the naïve brain microglia cells (82%) were found to be resting. CX3CR1+MHCII high cells were gated as activated microglia and tumor-bearing brain had a majority (75%) of activated microglia cells. Fig. S4 shows differences in microglial populations in various mouse tumor models.

We appreciate the reviewer for suggesting a great point about tumors being end-stage resulting in a higher frequency of infiltrating cells. A separate study on the kinetics of immune phenotypic changes as a function of tumor development would help address this hypothesis. Our GBM patient data (Fig. 5B) corroborates with the previously published study (Hambardzumyan,D et al, 2015) which reported microglia frequency to be between 30-50% in glioma patients. Previous studies (Korin et al 2017) has also shown presence of 70% microglia cells that are CD45 low, CD11b+ in naïve brain and 30% infiltrating CD45+ cells. Frequency differences could be attributed to the pre-run cell isolation technique. Our sample preparation also has a sizable proportion of CD11b-CD45- cells resulting in frequency differences. We have now included CyTOF and FACS comparison for T cell frequencies in supplementary Fig. S3.

Interestingly, authors do not see a difference in CD68+ macrophages in four different tumor types that have different origins and mutational profile. This is in contrast to many published works in the glioblastoma field, demonstrating that NF1 loss in human and murine models results in increased macrophage infiltration and correlated with high macrophage gene signature both in genetically engineered mouse models and MES GBM subtype. Authors should comment and discuss their findings and talk about differences with published literature.

Response: The reviewer has raised an excellent point. Higher CD68 expression has been associated with mesenchymal subtype of GBM characterized by NF1 loss and mutation. NF1 status of the GBM mouse models used in this study is not known. Furthermore, macrophages increase in numbers with progressive

stages of GBM with highest frequency of macrophages observed in stage-IV tumors. In our study, we have harvested tumors from end-stage humane end-point mice that might explain high albeit similar CD68+ frequencies in all mouse models.

Figure S6 has very low-quality IHC images and would benefit from sharper and clear staining. CD68 staining is inferior quality, and I am not sure how authors quantified the number of cells based on surface marker. I would advise to stain for IBA1 and quantify % positive area.

Response: We have now submitted better resolution images. The number of brown stained cells were manually counted in 1/4th of the image and numbers extrapolated for the whole image from there. We have also included Iba-1 in the IHC panel (Supplementary Fig. S10). To keep data analysis consistent for all the markers, we have counted brown cells as positive for Iba-1 staining as well.

Ly6C+ macrophages were classified as mononuclear MDSCs? Ly6C+ labels monocytes and, it is not a marker for macrophages. Newly infiltrated monocytes from circulation downregulate Ly6C upon differentiation into tissue macrophages. The authors should be cautious with terminology. And again, MDSC is a function of a cell and is not defined by Ly6C+CD11b+, as authors refer in the abstract. Without functional assay to show Ly6C+CD11b+ cells isolated from tumors can suppress function of T-cells, they should be called monocytes.

Response: We agree with the reviewer and as such have modified the terminology defining Ly6G+/Ly6C+ cells as Ly6G+ myeloid cells/neutrophils. Furthermore, Ly6G-/CD11b+/Ly6C+ cells have been renamed infiltrating monocytes while Ly6G-/Ly6C-/CD11b+ cells have been renamed as resident macrophages.

In the material and methods section, it states that Student's t-test was used, but there are instances that more than one group of comparison and that is not appropriate to use - Sfigure 4

Response: We have now modified the Materials and Methods section. Data were analyzed by Student t test with Holm-Sidak multiple corrections applied on graphpad prism. Data was expressed as mean \pm SEM and differences were considered significant at adjusted $P \leq 0.05$ (*) and $P \leq 0.01$ (**).

Figure 6 B is the 005 missing the error bars, or they are just that consistent? How many samples are included per tumor type?

Response: We have now revised the figure (previously Fig. 6B and now Fig. 5A and 5B) which now includes multiple repeats and more defined populations. N=3 for mouse tumors and N=5 for GBM patient samples.

Reviewer #3

Khalsa et al used a combination of H&E, IHC, RNAseq and CyTOF to perform comprehensive immune profiling on syngeneic mouse models of GBM. For each model, they additionally use CyTOF to compare the immune profiles between surgically resected and non-resected tumors and compare these results to those obtained from surgical resections from GBM patients. The findings represent an advance over what is currently known about how the immune response in syngeneic mouse models of GBM compares to that in human GBM, and the data presented represent an excellent resource for future studies. The major concern of this manuscript relates to a lack of statistical testing that is persistent throughout the results.

Major comments.

1. The statistical information throughout the manuscript is lacking, and related, the number of samples analyzed in the many comparisons is missing. Both are essential issues that need to be addressed. A non-comprehensive list of examples:

Response: We have now included the statistical information and number of samples in figure legends as well as methods section of the manuscript.

- Figure 1b, comparison of CD31 staining,

Response: We have now quantified CD31 staining and included the data as supplementary Figure S1D.

- 'showed dramatically lower Siglec F+ macrophage (CD11b+F4/80+Siglec F+MHCII+) and eosinophil (CD11b+Siglec F+MHCII lo) populations'

Response: After careful statistical analysis, we have revised the statement to "Mut 3 also had significantly fewer Siglec F+ macrophages (Siglec F+ CD11b+ MHCII+F4/80+)"

- 'CT2A tumor samples had more than two-fold higher monocytes/macrophages 215 compared to all the other tumor types'

Response: As the data pertaining to this figure was redundant, we have therefore removed it.

- The frequency differences outlined in Fig. 3B

Response: We have now performed detailed statistical analysis for this figure.

- The exhaustion marker analyses for T cells outlined in Supplementary Figure 6C. If the T cells do not significantly differ in their expression of exhaustion markers across models, this should be noted as it is valuable information for future experiments that use these models to study the effect of checkpoint inhibitors.

Response We thank the reviewers for pointing it out and have performed statistical testing on this figure. We evaluated CD4 and CD8 T cells for expression of various activation and inhibition markers (Supplementary Fig. S9A-S9B). GL261 tumors showed the lowest frequency of KLRG1+ cells and CD25+ cells of the CD4 subset (Supplementary Fig. S9A). Lag3 and Tim3 are markers of dysfunctional T cells and 70-80% of CD8 T cells in CT2A were positive for these markers (Supplementary Fig. S9B), significantly higher than the other tumor types. Also, the percentage of PD-1H+ CD4+ was higher than PD-1H+ CD8+ T cells in all tumor types as previously reported."

- There are no p-values for the analyses outlined in Supplementary Figure S8, despite claims that the number of T cells differs between subgroups of the models.

Response: We have now included statistical analysis to Supplementary Fig. S10 (previously Supplementary Fig. S8). We have also performed Student's T-test with Holm-Sidak corrections applied for multiple comparisons.

- There are no p-values for the CD8+ T cell-specific differences outlined in Figure 4B or Supplementary Figure S9. Data has been modified for these figures. There are no p-values for any of the immune cell differences outlined in Figure 5.

Response: We have now performed T-test correcting for multiple comparisons by Holm-Sidak method and included statistical significance information in Supplementary Fig. 11B and 11C (previously Fig. 4B) and Fig. 4E (previously Fig. S9).

There is no correlation coefficient demonstrating the similarity in immune profiles between patient GBMs and syngeneic mouse models.

Response: We performed Spearman's rank correlation coefficient test between each of the mouse model used and GBM patient data and found that 005 tumor had the highest value for R.

2. Are the differences in immune-cell related gene expression levels between specimens from the same model comparable, shown in Fig 1E, lesser or greater than the inter-model variability? For example, the differences between Mut3 and control seem negligible. Again, statistical testing is needed.

Response: The reviewer has raised an excellent point. In our studies, we tested this by performing principal component analysis to determine inter- and intra- group variability. PC1 separates naïve brain from all the tumor samples and PC2 separates Mut3 from rest of the tumors. Mut 3 and naïve brain samples cluster most distinctly as shown in Supplementary Fig. S2A. We also performed hierarchical cluster analysis in the heatmap of 100 most-distinct genes found in the RNA-sequencing data (supplementary Fig. S2B).

3. CT2A (and Mut3) tumor samples were classified as immunologically inert, GL261 (and 005) tumors as immune-active tumors. How do those observations compare with the better efficacy of CT2A tumors compared to GL261 when treated with autologous tumor lysates, reported in Genoud et al., *Oncoimmunology* 2018 (PMID 30524896).

Response: CT2A/GL261 comparison has been studied previously (Belmans, J. 2017) where pre-tumor treatment with autologous lysate was used as a subcutaneously delivered vaccine. In this Belmans, J. 2017 study, CT2A works better which could be due to differential antigen presentation. From our CyTOF data, we observe that CT2A tumors have comparatively more Ly6C+ macrophages known to carry soluble antigens from tissues to draining lymph nodes along with efferocytosis and cross-presentation of antigens (Larson et al 2016), CT2A also has fewer regulatory T cells than GL261 tumors that can explain a better anti-tumor response (Brezar, et al, 2016).

4. Immunologically inert tumors are suggested to contain fewer infiltrating immune cells, and this result seems at odds with the many reports that the frequency of innate myeloid immune cells is higher in mesenchymal glioblastomas, a category of GBM that is associated with worse outcomes compared to non-mesenchymal glioblastomas, (Wang et al, *Cancer Cell* 2017, PMID 28697342 & Engler, *PLOS One*, 2012, PMID 22937035).

Response: The reviewers have made a valid observation about myeloid cell frequency being highest in the mesenchymal tumors. These tumors are characterized by NF1 loss and mutation. NF1 status of the GBM mouse models used in this study is not known and thus, a direct subtyping of any of the mouse tumor models as mesenchymal is not valid. Further, mesenchymal GBMs are known to have more CD68+ cells (Wang et al, 2018) while our immunologically inert tumor types, containing fewer total infiltrating lymphocytes. Interestingly, the two mouse tumor models, CT2A and Mut3 have the fastest and slowest tumor progression, respectively, and thus, immunological inertness does not seem to be correlated with tumor progression.

5. Tumors were compared four days post resection; clarification is needed what this residual material represents since presumably there are few tumor cells left post-resection. Was any histology/immunohistochemistry performed to demonstrate presence of tumor cells post-resection? And what is the statistical significance of the findings?

Response: We have now added bioluminescent imaging pre and post resection in Fig. 4B to show that there is residual tumor left. In a majority of previous studies, we have shown that our bioluminescence imaging correlates with fluorescence imaging and the presence of fluorescent tumor cells post resection. We have also now added statistically different immune-cell populations in Fig. 4D as determined by T-test corrected for multiple comparisons (previously Fig. 5C).

6. Specimens from how many GBM patients were analyzed? Since GBMs from different expression subtypes have been shown to have very different immunological profiles (Wang et al, *Cancer Cell* 2017, PMID 28697342), the expression subtype of these GBMs needs to be reported to put findings into context.

Response: We have now added this information in the manuscript. Specifically, tissues from 5 WHO grade-IV glioma patients that were all IDH wild type were stained for CyTOF in this study.

7. The data outlined in the manuscript together represent an excellent resource for biologists to probe when planning experiments and need to be made accessible for exploration.

Response: We would like to thank the reviewer for considering our studies an excellent resource.

Minor comments:

1. Figure 1C: It would be helpful to have a box that indicates where in the 4X images the 20X images were taken.

Response: We have now addressed this in Fig. 1D (previously Fig. 1C).

2. The manuscript would benefit from a brief discussion of why each marker was chosen. What cell types were the author's hoping to find? What specific phenotypes were the authors searching for? Why were these markers the best markers for comparing each syngeneic model?

Response: Cell lineage defining markers were selected to cover most of the immune populations. CD45 is a pan-lymphocyte marker. T cell subsets can be identified by CD3, CD4, CD8 and TCRgd while NK cells can be defined by CD49b and NK1.1 (Arase, H, 2001). B cells express B220 and IgM is the most common isotype of B cells reported in tumors (Yuen, G, et al 2017). Various populations of innate immune cells including various monocyte/macrophage populations and dendritic cells can be identified by a combination of CD11b, CD11c, Siglec F, MHC-II, F4/80, CD103, CD115 and CD64 (Rei,Y, et al 2016, Rose S,et al 2012). Ly6c and Ly6g can help delineate various granulocytic populations (Rose S, et al 2012). Further, as activation/inhibition status is important for determining whether T cells present in the tumor micro-environment are functional, we added Tim3, Lag3, KLRG1, PD-1H, CD25, CD44, CD62L and CD86 (Xia, A, et al 2019). CD39 is an ectonuclease that serves as an inhibitory marker on cell surface of many different immune cell populations (Canale, F, 2018, Raczowski, et al,2018). We have included this in methods section of our manuscript.

3. The asterisks indicating significance should be outlined in each figure legend rather than in the methods section.

Response: We would like to thank the reviewer for pointing this out. We have now addressed this in each figure legend.

4. What is the difference between a classical CD8+ T cell (which the authors also refer to as functional) and a tumor-reactive CD8+ T cells?

Response: Tumor reactive CD39+ CD103+ CD8+ T cells have been previously identified in ovarian, rectal, melanoma and non-small cell lung cancer patients and correlated with increased overall survival. We found this population in our syngeneic mouse models with the highest frequency being present in the GL261 mouse model. Implication and function of these CD39+CD103+ CD8 T cells in the context of GBM in mouse models and patients of GBM needs to be analyzed". We have discussed this in the discussion section of our manuscript.

Figure A: High exposure of pAKT/AKT blot for tumor lines.

REVIEWER COMMENTS

Reviewer #1 (Remarks to the Author):

The authors have addressed my main points or at least acknowledged the rationale for their designs and data interpretation.

Reviewer #2 (Remarks to the Author):

Authors addressed all my comments, manuscript is much stronger and clear.

Reviewer #3 (Remarks to the Author):

The revision by Khalsa et al has addressed some issues previously raised. Several key criticisms were not or poorly considered.

Major

1. There are statements in the manuscript that are subjective interpretations, whereas quantification of the data and statistical testing of the numbers would make these same statements objective. These are just a few examples as there are too many statements in the paper that are void of any evidence:
 - a. "Mice implanted with CT2A tumors took the shortest time to reach end-stage while tumor progression of Mut 3 was the slowest (Fig. 1C)." A logrank test is required to implicate that there is significant slower growth. Also, "Kaplan-Meier" is misspelled in the written legend.
 - b. Figure S2C shows "hairballs" of connected terms based on differentially expressed genes. It is unclear if these genes were upregulated or downregulated, how they were defined, what level of statistical significance was used, or how they were related to the terms that are being shown.
 - c. Visually, Figure 1E shows that CT2A is much more closely related to GL261 and 005. However, it is suggested that it bears similarity to Mut3.
 - d. Tumors showed highest expression of T cell activation related genes whereas GL261 showed higher expression level of antigen processing and presentation related genes than 005 tumors
2. The associated datasets need to be publicly available, in accordance with the publishing guidelines of this journal (<https://www.nature.com/nature-research/editorial-policies/reporting-standards#availability-of-data>).
3. The previous concern that CT2A was classified as immunologically inert, and GL261 as immune-active tumors, and how this compares to results reported in Genoud et al., *Oncoimmunology* 2018 (PMID 30524896), was not addressed. Several additional citations were discussed but the issue at hand remains in the manuscript.
4. GBM subtype of patient specimens is of key importance to be able to interpret the results and is still not reported.

Minor

1. Not sure what the following sentence is trying to say: "However, unlike other tumor types, immune cell number is small and isolation is challenging making it difficult to assess immune components by routine flow cytometry with very few studies performed on Tumor infiltrating immune cells (TIIC) isolated from patients."
2. The English grammar could use some improvement.
3. Consider adding the word 'murine' or 'mouse' to the title for improved recognition.

Response to Reviewers

Reviewer #3 (Remarks to the Author):

The revision by Khalsa et al has addressed some issues previously raised. Several key criticisms were not or poorly considered.

Major

1. There are statements in the manuscript that are subjective interpretations, whereas quantification of the data and statistical testing of the numbers would make these same statements objective. These are just a few examples as there are too many statements in the paper that are void of any evidence:

a. "Mice implanted with CT2A tumors took the shortest time to reach end-stage while tumor progression of Mut 3 was the slowest (Fig. 1C)." A logrank test is required to implicate that there is significant slower growth. Also, "Kaplan-Meier" is misspelled in the written legend.

Response: We would like to thank the reviewer for pointing this out. We have now performed the log rank test which indicates significant differences between different tumor types. As pointed out by the reviewer, we have also corrected the spelling.

b. Figure S2C shows "hairballs" of connected terms based on differentially expressed genes. It is unclear if these genes were upregulated or downregulated, how they were defined, what level of statistical significance was used, or how they were related to the terms that are being shown.

Response: We have changed the data representation to dot plots in Supplementary Fig. S2C (See revised Supplementary Fig. S3 and S4: also included in this letter). These plots also show p-adjusted values and the data indicates pathways that were differentially enriched between naïve brain tissue and each of the tumor-bearing brain tissue. These were defined by first identifying differentially expressed genes (both up- and down- regulated) for each tumor type using naïve brain as reference followed by Gene ontology analysis that gave the pathways that are enriched for these differentially expressed genes. Chi square analysis was performed with a false discovery rate cut off of 0.05 and were corrected for multiple comparisons using Benjamini Hochberg method.

c. Visually, Figure 1E shows that CT2A is much more closely related to GL261 and 005. However, it is suggested that it bears similarity to Mut3.

Response: Figure 1E shows log-fold expression levels of immune-related pathways ranging from 1 (white) to 25 (bright orange) as can be seen in the scale. CT2A and Mut 3 are all white as compared to GL261 and 005.

d. Tumors showed highest expression of T cell activation related genes whereas GL261 showed higher expression level of antigen processing and presentation related genes than 005 tumors

Response: We have now deleted this statement and also modified/deleted additional statements in the manuscript that lack evidence.

2. The associated datasets need to be publicly available, in accordance with the publishing guidelines of this journal (<https://www.nature.com/nature-research/editorial-policies/reporting-standards#availability-of-data>).

Response: We are currently working on uploading CyTOF data on ImmPort and RNA sequencing data on GEO database. We will share the accession numbers as soon as they are ready.

3. The previous concern that CT2A was classified as immunologically inert, and GL261 as immune-active tumors, and how this compares to results reported in Genoud et al., Oncoimmunology 2018 (PMID

30524896), was not addressed. Several additional citations were discussed but the issue at hand remains in the manuscript.

Genoud et al, 2018 study titled "Responsiveness to anti-PD-1 and anti-CTLA-4 immune checkpoint blockade in SB28 and GL261 mouse glioma models" compared the efficacy of immune checkpoint inhibitor, CTLA-4 in SB28 and GL261 tumor models. This study reported the unsuitability of GL261 model for immune-checkpoint inhibitor therapy as 50% of GL261 tumor bearing mice got cured. On the contrary, SB28 was less immunogenic and more resistant to therapy which is more reflective of GBM tumors in patients. Given the similarity of low T cell infiltration in SB28 and CT2A, we believe that CT2A tumor model used in our studies (see Fig. S12) is similar to SB28 mouse model used in Genoud et al studies.

The study comparing CT2A and GL261 tumor model (Belmans, J. 2017) utilized pre-tumor treatment with autologous lysate as a subcutaneously delivered vaccine. In this study, CT2A tumors have a better response which could be due to differential antigen presentation. Our CyTOF data indicate that CT2A tumors have comparatively more Ly6C+ macrophages known to cargo soluble antigens from tissues to draining lymph nodes along with efferocytosis and cross-presentation of antigens (Larson et al 2016), CT2A also has fewer regulatory T cells than GL261 tumors that can explain a better anti-tumor response (Brezar, et al, 2016). Based on these findings, we have now revised the discussion:

"Although GL261 is widely used in pre-clinical studies, these tumor-bearing mice have significantly fewer APCs and more T cells than GBM patients that are usually suppressive in nature. This immune-phenotype could be a factor contributing to better efficacy of treatment modalities that target T cell populations in GL261 in comparison to other mouse models (Genoud et al, 2018). In this study, we report a plethora of immune-phenotypic data that can be utilized to make better choices of the mouse model to be used for testing various anti-tumor therapies. We report higher frequency of Ly6C+ macrophages in CT2A that mediate efferocytosis and cross presentation of antigens that can explain why CT2A responds better to vaccination with autologous lysate, as compared to GL261 tumors (Belmans J, 2017)."

4. GBM subtype of patient specimens is of key importance to be able to interpret the results and is still not reported.

Previous studies have classified GBM tumors as proneural, mesenchymal and classical based on expression of selected genetic events such as PDGFRA/IDH1, NF1 and EGFR alterations (Verhaak, R. et al, 2010). However, recent studies based on single cell RNA sequencing and multi-region sampling have shown tumor cell heterogeneity in expression profile within the same tumor (Nefel, C. et al 2019, PMID: 31327527, Patel, AP. et al, 2014, PMID: 24925914). Longitudinal analysis have also revealed that GBM subtypes can change over time (Wang, Q. et al 2017, PMID: 28697342). In line with these findings, our institution relies on currently used genetic markers with better prognostic value: MGMT and IDH (Weller, M. et al, 2011 PMID: 21607882, Staedtke, . et al, 2016, PMID: 28603776), and therefore doesn't classify GBM patients regularly as proneural, mesenchymal and classical.

All GBM patient samples used in the study were WHO grade IV Glioblastoma that were IDH1 WT with a mix of MGMT methylated and unmethylated status. Two patients had EGFR amplification, 4 had PTEN loss, 3 patients showed CDKN2A loss whereas 1 showed TP53 loss on genetic screening. However, we did not exclude any patients from the study based on genetic subtyping. This information has now been included in the methods section of the manuscript.

Minor

1. Not sure what the following sentence is trying to say: "However, unlike other tumor types, immune cell number is small and isolation is challenging making it difficult to assess immune components by routine flow cytometry with very few studies performed on Tumor infiltrating immune cells (TIIC) isolated from patients."

Response: This comment is to explain the importance of CyTOF specifically in immune-phenotyping of brain tumors that have very few tumor infiltrating immune cells in comparison to other tumor types such as breast cancer. We have now revised the sentence as “Studies involving isolation of immune cells from the brain and flow cytometry-based evaluation of brain-infiltrating immune cells present unique challenges such as poor viability and high autofluorescence. These challenges were overcome only recently for brain tissue with better isolation protocols and evolution of flow cytometry to study up to 21 markers simultaneously.”

2. The English grammar could use some improvement.

Response: We have corrected the grammatical errors in the manuscript.

3. Consider adding the word ‘murine’ or ‘mouse’ to the title for improved recognition.

Response: We have now modified the title to “Immune phenotyping of diverse syngeneic murine brain tumors identifies immunologically distinct types”